# INDUCTIVE LOTTERY TICKET LEARNING FOR GRAPH NEURAL NETWORKS

## ABSTRACT

Deep graph neural networks (GNNs) have gained increasing popularity, while usually suffer from unaffordable computations for real-world large-scale applications. Hence, pruning GNNs is of great need but largely unexplored. A recent work (Chen et al., 2021) studies lottery ticket learning for GNNs, aiming to find a subset of model parameters and graph structure that can best maintain the GNN performance. However, it is tailed for the transductive setting, failing to generalize to unseen graphs, which are common in inductive tasks like graph classification. In this work, we propose a simple and effective learning paradigm, Inductive Co-Pruning of GNNs (ICPG), to endow graph lottery tickets with inductive pruning capacity. To prune the input graphs, we design a predictive model to predict importance scores for each edge based on the input; to prune the model parameters, it views the weight's magnitude as their importance scores. Then we design an iterative co-pruning strategy to trim the graph edges and GNN weights based on their importance scores. Although it might be strikingly simple, ICPG surpasses the existing pruning method and can be universally applicable in both inductive and transductive learning settings. On ten graph-classification and two node-classification benchmarks, ICPG achieves the same performance level with 14.26%∼43.12% sparsity for graphs and 48.80%∼91.41% sparsity for the model.

## 1 INTRODUCTION

Graph neural networks (GNNs) (Kipf & Welling, 2016; Veličković et al., 2017; Zhou et al., 2018) have become a prevalent solution for machine learning tasks on graph-structured data. Such success is usually ascribed to the powerful representation learning of GNN, which incorporates the graph structure into the representations, such as aggregating neural messages from the neighboring nodes to update the ego node's representation (Veličković et al., 2017; Kipf & Welling, 2016).

As the field grows, there is an increasing need of building deeper GNN architectures (Li et al., 2020a; 2021) on larger-scale graphs (Hu et al., 2020). While deepening GNNs shows potentials on large-scale graphs, it also brings expensive computations due to the increased scale of graph data and model parameters, limiting their deployment in resource-constrained applications. Taking fraud detection in a transaction network as an example, the scale of user nodes easily reaches millions or even larger, making a GNN-detector model prohibitive to stack deep layers and predict the malicious behaviors in real time. Hence, pruning over-parameterized GNNs is of great need, which aims to answer the question: Can we co-sparsify the input graphs and model parameters, while preserving or even improving the performance?

Recently, the pruning approach for GNNs, UGS, (Chen et al., 2021), is proposed to find graph lottery tickets (GLTs) — smaller subsets of model parameters and input graphs. At its core is Lottery Ticket Hypothesis (LTH) (Frankle & Carbin, 2018) speculating that any dense, randomly-initialized neural network contains a sparse subnetwork, which can be trained independently to achieve a matching performance as the dense network. Specifically, UGS employs trainable masks on each edge in the input graph and each weight in the model parameters, to specify their importance. When training the model with the masks, the strategy of iterative magnitude-based pruning (IMP) (Frankle & Carbin, 2018) is used to discard the edges and weights with the lowest mask values at each iteration.

Despite effectiveness, there exist the following limitations: (1) UGS focuses solely on providing the **transductive** graph masks by generating a painstakingly customized mask for a single edge

individually and independently. That is, the edge masks are limited to the given graph, making UGS infeasible to be applied in the **inductive** setting, since the edge masks hardly generalize to unseen edges or entirely new graphs. (2) Applying a mask for each edge alone only provides a **local** understanding of the edge, rather than the **global** view of the entire graph (*e.g.,* in node classification) or multiple graphs (*e.g.,* in graph classification). Moreover, the way of creating trainable edge masks will double the parameters of GNNs, which violates the purpose of pruning somehow. As a result, these edge masks could be suboptimal to guide the pruning. (3) The unsatisfactory graph pruning will negatively influence the pruning of model weights. Worse still, low-quality weight pruning will amplify the misleading signal of edge masks in turn. They influence each other and form a vicious circle. We ascribe all these limitations of UGS to its transductive nature. Hence, conducting the combinatorial pruning in the inductive setting is crucial to high-quality winning tickets.

In this work, we emphasize the inductive nature within the combinatorial pruning of input graphs and GNN parameters and present our framework, Inductive Co-Pruning of GNNs (**ICPG**). It is an extremely simple but effective pruning framework that is applicable to any GNN in both inductive and transductive settings. Specifically, for the input graphs, we design a generative probabilistic model (termed AutoMasker), which learns to generate edge masks from the observed graphs. It is parameterized with an additional GNN-based encoder, whose parameters are shared across the population of observed graphs. As a consequence, AutoMasker is naturally capable to specify the significance for each edge and extract core-subgraphs from a global view of the entire observations. For the model parameters, we simply exploit the magnitude of a model weight to assess whether it should be pruned, rather than training an additional mask (Chen et al., 2021). Having established the edge masks and weight magnitudes, we can obtain high-quality GLTs by pruning the lowest-mask edges and lowest-magnitude weights. Experiments on ten graph-classification and two node-classification benchmarks consistently validate our framework ICPG by identifying high-quality GLTs. The visualizations show that ICPG always retains decisive subgraphs, such as edges located on digital pixels in MNIST graphs, which further illustrates the effectiveness and rationality. Moreover, we inspect the graph- and GNN-level transferability of GLTs, which promises for deploying our ICPG in the pre-training and fine-tuning paradigm. Our main contributions can be summarized as follows:

- We proposed a simple but effective pruning framework, ICPG, to prune the GNN model and input graphs simultaneously. It can identify high-quality GLTs in diverse tasks of graph representation learning, under both inductive and transductive settings.

- For graph classification tasks, ICPG can locate GLTs from small-scale (TUDataset), medium-scale (Superpixel graphs) to challenging large-scale datasets (OGB) with graph sparsity from 22.62%~43.12% and GNN sparsity from 67.23%~91.41% with no degradation on performance.

- For node classification tasks, ICPG can effectively develop on both transductive learning (Cora dataset) and inductive learning (PPI dataset), which identifies the GLTs with graph sparsity from 22.62%~26.49% and GNN sparsity from 67.23%~73.79% without sacrificing performance.

- The proposed AutoMasker promises for both GNN-level and graph-level transferability, which can achieve comparable or even better performance as compared with the original full graphs. In-depth analyses with visual inspections further demonstrate the effectiveness and rationality.

## 2 RELATED WORK

**Graph Neural Networks** (GNNs) (Kipf & Welling, 2016; Veličković et al., 2017; Xu et al., 2019; Ying et al., 2018) have emerged as a powerful tool for learning the representation of graph-structured data. The great success mainly comes from the structure-aware learning, which follows the iterative message-passing scheme (Veličković et al., 2017). Specifically, we denote an undirected graph by $\mathcal{G} = (\mathbf{A}, \mathbf{X})$ with the node set $\mathcal{V}$ and edge set $\mathcal{E}$. $\mathbf{A} \in \{0,1\}^{|\mathcal{V}| \times |\mathcal{V}|}$ is the adjacency matrix, where $\mathbf{A}[i,j] = 1$ denotes the edge between node $v_i$ and node $v_j$, otherwise $\mathbf{A}[i,j] = 0$. $\mathbf{X} \in \mathbb{R}^{|\mathcal{V}| \times d}$ is the matrix of node features, where $\boldsymbol{x}_i = \mathbf{X}[i,:]$ is the $d$-dimensional feature of the node $v_i \in \mathcal{V}$. Given a $K$-layer GNN, its $k$-th layer generates the representation of node $v_i$ as:

$$\boldsymbol{a}_i^{(k)} = \text{AGGREGATION}^{(k)}(\{\boldsymbol{h}_j^{(k-1)}|j \in \mathcal{N}(i)\}), \boldsymbol{h}_i^{(k)} = \text{COMBINE}^{(k)}(\boldsymbol{h}_i^{(k-1)}, \boldsymbol{a}_i^{(k)}), \quad (1)$$

where $\boldsymbol{h}_i^{(k)}$ and $\boldsymbol{a}_i^{(k)}$ are the representation of node $v_i$ and the message aggregated from its neighbor nodes set $\mathcal{N}(i)$, respectively; the AGGREGATION and COMBINE operators are the message

passing and update functions, respectively. After propagating through $K$ layers, we get the final representations of nodes, which facilitate downstream node-level tasks, such as node classification and link prediction. As for graph-level tasks like graph classification and graph matching, we further hire the READOUT function to generate the representation of the whole graph $\mathcal{G}$:

$$\boldsymbol{Z}_{\mathcal{G}} = \text{READOUT}(\{\boldsymbol{h}_i^{(k)} | v_i \in \mathcal{V}, k \in \{1, \cdots, K\}\}). \quad (2)$$

Various GNNs, such as GIN (Xu et al., 2019) and GAT (Veličković et al., 2017), implement different AGGREGATION, COMBINE and READOUT functions, so as to refine the desired information from graph structures.

**Graph Sparsification or Sampling** (Voudigari et al., 2016; Leskovec & Faloutsos, 2006) aims to find small core-subgraphs from the original graph, which can remain effective for graph learning tasks. Numerous strategies (Zeng et al., 2019; Franceschi et al., 2019; Ying et al., 2019; Hamilton et al., 2017; Chen et al., 2018) are proposed to achieve efficient graph learning. For example, GraphSAGE (Hamilton et al., 2017) samples and aggregates feature from a node's local neighborhood. FastGCN (Chen et al., 2018) adopts the global importance sampling, which is more efficient for training. DropEdge (Rong et al., 2019) randomly drops edges from the input graph, which can be seen as a data augmenter. Another research line selects the core-subgraph in an optimization way. SGCN (Li et al., 2020b) adopts the ADMM optimization algorithm to sparsify the adjacency matrix. UGS (Chen et al., 2021) utilizes a trainable mask for each edge to remove the potential task-irrelevant edges. Distinct from them, our AutoMasker predicts the importance score of each edge from a global view, thus having better generalization ability in the inductive settings.

**Lottery Ticket Hypothesis** (LTH) (Frankle & Carbin, 2018) states that a sparse subnetwork exists in a dense randomly-initialized network that can be trained to achieve comparable performance to the full models. LTH is explored in many fields such as computer vision and natural language processing (Chen et al., 2020; Ma et al., 2021; Yin et al., 2020; Liu et al., 2019; Wang et al., 2020; Savarese et al., 2020). Recently, Chen et al. (2021) extends the LTH to the GNNs and proposes the Graph Lottery Ticket (GLT), which includes subgraph and subnetwork pairs that can be trained independently to reach comparable performance to the dense pairs. However, due to the transductive nature of graph-specific masks, UGS (Chen et al., 2021) can not develop on inductive learning settings, such as graph classification tasks. To tackle the dilemma, we propose AutoMasker to globally learn the significance of each edge from training graphs and predict importance scores for new coming graphs, which is graph-independent and inductive.

## 3 METHODOLOGY

Here we first formulate the task of learning graph lottery tickets, and then present our inductive strategy of co-pruning the input graphs and model weights.

### 3.1 INDUCTIVE GRAPH LOTTERY TICKET

Without loss of generality, we consider the inductive task of graph classification as an example. Given a GNN classifier $f(\cdot, \boldsymbol{\Theta}_{g_0})$, it starts from the randomly-initialized parameters $\boldsymbol{\Theta}_{g_0}$ before training and arrives at the well-optimized parameters $\boldsymbol{\Theta}_g$ after training. Once trained, it takes any graph $\mathcal{G} = (\mathbf{A}, \mathbf{X})$ as the input and yields a probability distribution over $C$ classes $\hat{\mathbf{y}} = f(\mathcal{G}, \boldsymbol{\Theta}_g)$. Wherein, $\mathcal{G}$ is associated with the adjacency matrix $\mathbf{A}$ and the pre-existing node features $\mathbf{X}$.

The goal of learning graph lottery tickets is to make the input graph $\mathcal{G}$ and the model weights $\boldsymbol{\Theta}_{g_0}$ sparse to reduce the computational costs, while preserving the classification performance. Formally, it aims to generate two masks $\mathbf{m}_{\mathcal{G}}$ and $\mathbf{m}_{\Theta}$, which are applied on $\mathcal{G}$ and $\boldsymbol{\Theta}_{g_0}$ correspondingly, so as to establish the sparser input graph $\mathcal{G}' = (\mathbf{m}_{\mathcal{G}} \odot \mathbf{A}, \mathbf{X})$ and initialized weights $\boldsymbol{\Theta}'_{g_0} = \mathbf{m}_{\Theta} \odot \boldsymbol{\Theta}_{g_0}$. Hereafter, through retraining the subnetwork $f(\cdot, \boldsymbol{\Theta}'_{g_0})$ on the sparse versions $\{\mathcal{G}'\}$ of training graphs, we can get the new converged parameters $\boldsymbol{\Theta}'_g$. If the well-optimized subnetwork can achieve comparable performance with full graphs and network, we term the pair of $\mathcal{G}'$ and $f(\cdot, \boldsymbol{\Theta}'_{g_0})$ as graph lottery tickets (GLTs).

Although a very recent study, UGS (Chen et al., 2021), has proposed an approach to learn the GLTs, it focuses solely on the transductive setting but leaves the inductive setting untouched. Specifically,

it assigns a trainable mask to each edge of the input graph and trains such graph-specific masks individually and independently. As a consequence, these edge-dependent masks are limited to the given graph, hardly generalizing to unseen edges or entirely new graphs. Distinct from UGS, we aim to uncover GLTs in inductive learning settings.

## 3.2 AUTOMASKER

Instead of assigning mask to single edge, our idea is extremely simple: we take a collection of graph instances and design a trainable model to learn to mask edges collectively. The key ingredient towards this model is an additional GNN-based model, termed AutoMasker, whose parameters are shared across the population of observed graphs. Here we represent AutoMasker as the combination of a graph encoder and a subsequent scoring function. Formally, given a graph $\mathcal{G} = (\mathbf{A}, \mathbf{X})$, AutoMasker applies a GNN-based graph encoder $g(\cdot)$ to create representations of all nodes as:

$$\mathbf{H} = g(\mathbf{A}, \mathbf{X}), \tag{3}$$

where $\mathbf{H} \in \mathbb{R}^{|\mathcal{V} \times d|}$ stores $d$-dimension representations of all nodes, whose $i$-th row $\mathbf{h}_i$ represents the representation of node $v_i$; $g(\cdot)$ is a GNN following the message-passing paradigm in Equation 1. To assess the importance score of edge $(i, j)$ between node $v_i$ and node $v_j$, AutoMasker builds a multi-layer perceptron (MLP) upon the concatenation of node representations $\mathbf{h}_i$ and $\mathbf{h}_j$, which yields the score $\alpha_{ij}$. In what follows, the sigmoid function $\sigma(\cdot)$ projects $\alpha_{ij}$ into the range of $(0, 1)$, which represents the probability of edge $(i, j)$ being the winning ticket. The scoring function is represented as follows:

$$s_{ij} = \sigma(\alpha_{ij}), \quad \alpha_{ij} = \text{MLP}([\mathbf{h}_i, \mathbf{h}_j]). \tag{4}$$

By employing the scoring function over all possible edges, we are able to collect the matrix of edge masks $\mathbf{s}_{\mathcal{G}}$, where $\mathbf{s}_{\mathcal{G}}[i, j] = s_{ij}$ if edge $(i, j)$ holds, otherwise $\mathbf{s}_{\mathcal{G}}[i, j] = 0$. In a nutshell, we summarize the AutoMasker function as follows:

$$\mathbf{s}_{\mathcal{G}} = \text{AutoMasker}(\mathcal{G}, \mathbf{\Theta}_a), \tag{5}$$

where $\mathbf{\Theta}_a$ is the trainable parameters of AutoMasker, covering the parameters of the GNN encoder and the MLP.

Although the key ingredient of AutoMasker is simple, it has several conceptual advantages over UGS: (1) **Global view:** Although edge masks derived from UGS might preserve the fidelity to local importance, they do not help to delineate the general picture of the whole graph population. Distinct from UGS, our AutoMasker takes a global view of the graph population, which enables us to identify the edge coalitions. Specifically, as edges usually collaborate with each other to make predictions, rather than working individually, they form a coalition like the functional groups of a molecule graph, the community of a social network. Considering such coalition effects, AutoMasker is able to measure the importance of edges more accurately. (2) **Lightweight edge masks:** When using UGS to prune graph data with millions of edges or nodes, the cost of assigning local edge masks one-by-one will be prohibitive with such a large-scale dataset in real-world scenarios. Moreover, UGS introduces additional parameters, whose scale remains the same as the edge numbers $\sum_{\mathcal{G}} |\mathcal{E}|$ and is much larger than the original parameters being pruned. Hence, it somehow violates the purpose of pruning. In our AutoMasker, the additional parameters are *i.e.,* $\mathbf{\Theta}_a$ in Equation 5 only and remain invariant across the change of data scale. (3) **Generalization:** AutoMasker can generalize the mechanism of mask generation to new graphs without retraining, making it more efficient to prune unseen and large-scale graphs.

## 3.3 INDUCTIVE CO-PRUNING STRATEGY

Here we present the framework of Inductive Co-Pruning of GNNs (ICPG) to localize GLTs inductively. Figure 1 demonstrates its overview, which consists of the following two steps:

- **Step 1: Co-training AutoMasker and the GNN model of interest.** Given an input graph $\mathcal{G} = (\mathbf{A}, \mathbf{X})$, AutoMasker first generates the edge mask $\mathbf{s}_{\mathcal{G}}$ via Equation 5. Then we apply $\mathbf{s}_{\mathcal{G}}$ to the adjacency matrix $\mathbf{A}$ to create the masked graph $\mathcal{G}_s = (\mathbf{s}_{\mathcal{G}} \odot \mathbf{A}, \mathbf{X})$, which fully reflects AutoMasker's decision for the importance of each edge, such that less important edges are prone to have lower mask values. Finally, we feed the masked graphs into the GNN model to co-train the

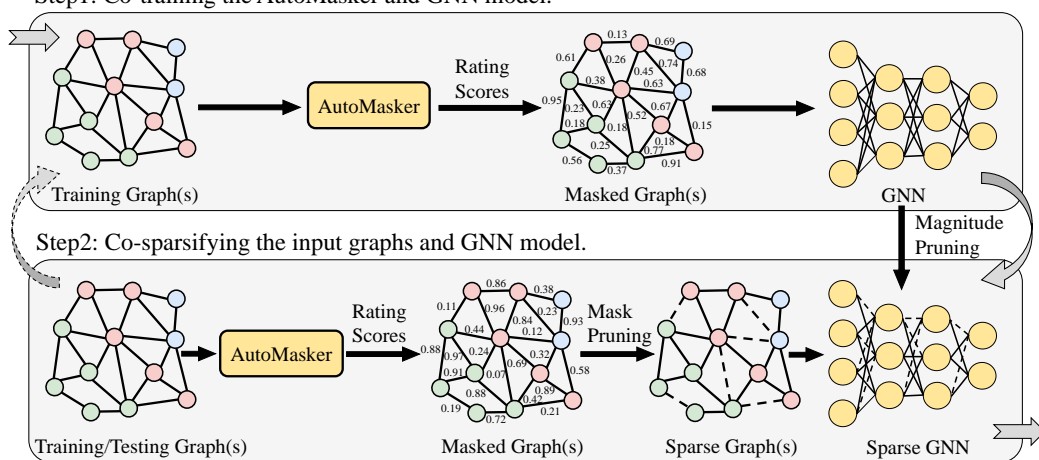

Figure 1: The Inductive Co-Pruning of GNNs (ICPG) framework to find the GLTs.

AutoMasker and the model. The GNN model adopts the masked graph to learn the representation and make predictions, which can be viewed as the supervision signals to guide the AutoMasker to achieve a more accurate decision. The detailed co-training process is shown in Algorithm 1 of Appendix A1.1. When the training is done, we conduct **Step 2** to perform the pruning.

- **Step 2: Co-sparsifying the input graphs and GNN model.** Having obtained the well-trained AutoMasker and GNN model, we can apply the knowledge learned from numerous graphs to co-sparsify the graphs and the model. For graphs, we adopt AutoMasker to predict the importance of all the edges for each graph. Then the edges of a certain graph are sorted based on the mask values, and the edges with 5% the lowest values are pruned to obtain the binary graph mask $\mathbf{m}_{\mathcal{G}}$. For GNN, we sort the parameters based on the weight magnitude and prune 20% the lowest-magnitude parameters to obtain the binary model mask $\mathbf{m}_{\Theta}$. Under the current sparsity level, we now successfully obtain the sparsified graph $\mathcal{G}' = (\mathbf{m}_{\mathcal{G}} \odot \mathbf{A}, \mathbf{X})$ and the sparsified mask $\mathbf{m}_{\Theta}$ for the model. Finally, we need to check whether the sparsity meets our condition. If the sparsity is satisfied, the algorithm is completed; if not, we need to reuse the found GLT to update the original graphs and GNN model, and iteratively use **Step 1** and **Step 2** (dotted arrow in Figure 1) until the condition is met. In Appendix A1.1, Algorithm 2 offers the detailed algorithm of ICPG.

## 4 EXPERIMENTS

In this section, we conduct extensive experiments on diverse benchmarks to validate the effectiveness of the ICPG. We first introduce the experimental settings in Section 4.1, and explore the existence of GLTs in graph classification and node classification in Sections 4.2 and 4.3, respectively. We also validate the transferability of the AutoMasker in Section 4.4. More ablation studies and visualizations are provided in Sections 4.5 and 4.6, respectively.

### 4.1 EXPERIMENTAL SETTINGS

**Datasets.** For graph classification, we use two biological graphs (NCI1, MUTAG), four social graphs (COLLAB, RED-B, RED-M5K, RED-M12K) (Morris et al., 2020), two superpixel graphs (MNIST, CIFAR-10) (Knyazev et al., 2019), and two large-scale Open Graph Benchmark (ogbg-ppa and ogbg-code2) (Hu et al., 2020). For node classification, we choose the transductive learning dataset: Cora, and the inductive learning dataset: PPI. More details are provided in Appendix A1.

**Models and Training Details.** We adopt the same architecture for the GNN model and GNN encoder for AutoMasker. For all graph classification datasets and Cora dataset, we adopt the GCN (Kipf & Welling, 2016) model with different layers and hiddens. For the PPI dataset, we choose the GAT (Veličković et al., 2017) network to achieve a better baseline performance as work (Veličković et al., 2017). More details about models and training are provided in Appendix A1.4 and A1.5.

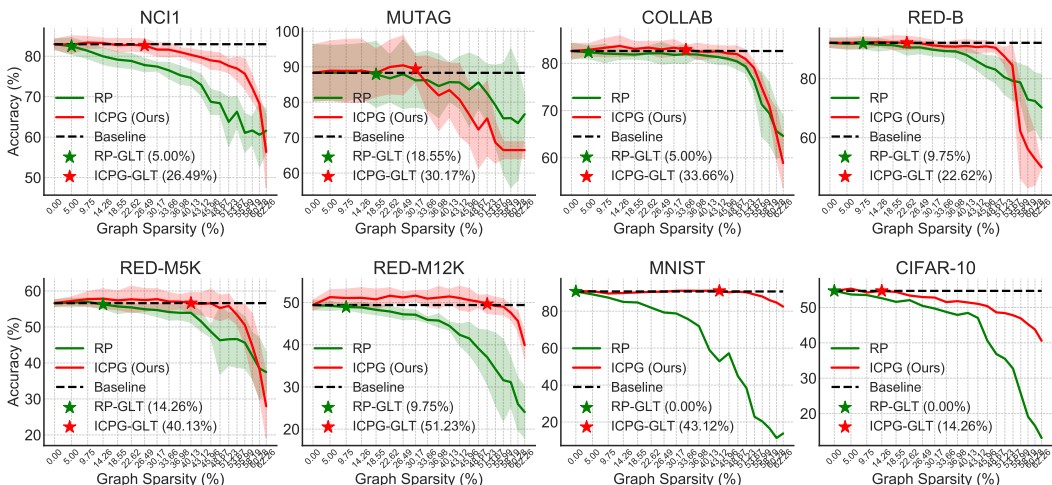

Figure 2: Graph classification performance on achieved graph sparsity.

## 4.2 GRAPH LOTTERY TICKETS IN GRAPH CLASSIFICATION

We first conduct experiments to find the GLTs in graph classification tasks. The results are displayed in Figure 2 and Figure 3. We also plot the random pruning (RP) for better comparison. Stars denote the extreme sparsity, which is the maximal sparsity-level without performance degradation. More results about GNNs sparsity are provided in Appendix A2.1. We make the following **Obs**ervations:

**Obs.1. GLTs extensively exist in graph classification tasks.** Utilizing the ICPG, we can successfully locate the GLTs with different sparsity-levels from different types of graphs. For NCI1 and MUTAG, we precisely identify GLTs with the extreme graph sparsity at 26.49% and 30.17%, GNN sparsity at 73.79% and 79.03%, respectively. On four social network datasets, we find the GLTs with graph sparsity of 22.62%~51.23% and GNN sparsity-level in 67.23%~95.60%. For MNIST and CIFAR-10, the GLTs are achieved with graphs sparsity of 43.13% and 14.26%, GNN sparsity of 91.41% and 48.80%. These results show that ICPG can locate the high-quality GLTs in graph classification tasks with different graph types, and demonstrate the potential of efficient training or inference with sparser graphs and lightweight GNNs without sacrificing performance.

**Obs.2. AutoMasker has good generalization ability.** The mainstream graph sparsification techniques (Chen et al., 2021; Zheng et al., 2020; Li et al., 2020b) cannot inductively prune unseen graphs. However, the AutoMasker can flexibly overcome this challenge. Compared with random pruning (RP), our proposed ICPG can find more sparse subgraphs and subnetworks and keep a large gap with RP. For instance, the RED-M5K and RED-M12K graphs pruned by ICPG can achieve 40.13% and 51.23% extreme graph sparsity, improving 25.87% and 41.48% compared with RP, which keeps an extremely large superiority. The excellent results indicate that AutoMasker can precisely capture more significant core-patterns from the training graphs and have a good generalization ability to predict the high-quality masks for unseen graphs.

**Obs.3. The extreme sparsity of GLTs depends on the property of graphs.** Although ICPG achieves higher sparsity than RP on most graphs, the improvements are not obvious on a small part of the graphs, such as biochemical molecule graphs: NCI1 or MUTAG. We make the following conjectures: Firstly, most of the edges in these graphs are important, such as a certain edge may correspond to a crucial chemical bond, which may drastically affect the chemical properties of the molecule if pruned.

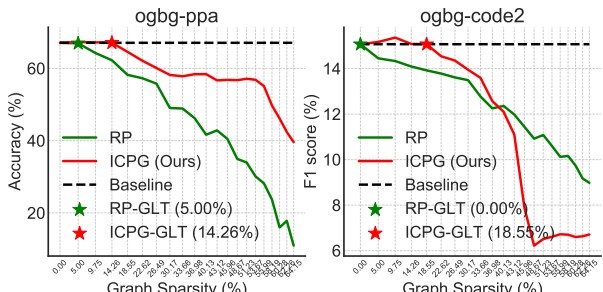

Figure 3: Graph classification performance over achieved graph sparsity on large-scale datasets.

Secondly, the graph size is relatively small, which just includes a few dozen nodes and edges, so it is more sensitive to pruning. On the contrary, the larger social network datasets, such as RED-M5K

or RED-M12K, contain hundreds or thousands of edges for each graph, so there may be plenty of redundant edges that demonstrate the insensitivity to pruning.

**Obs.4. AutoMasker can well tackle larger-size and larger-quantity graphs.** Figure 3 demonstrates the results on the challenging OGB datasets, which are consist of larger-size graphs (2266.1 edges and 243.4 nodes on average per graph for ogbg-ppa) and larger-quantity graphs (452,741 graphs for ogbg-code2). We surprisingly find the OGB datasets are so intractable that RP can only locate 5% graph sparsity of GLT on the ogbg-ppa, and it is even impossible to find any sparser GLTs on the ogbg-code2. Despite this, the proposed ICPG can locate the GLTs with 14.26% and 18.55% graph sparsity, 48.80% and 59.40% GNN sparsity on ogbg-ppa and ogbg-code2, respectively. The superior performance further verifies the strong scalability and generalization of the AutoMasker.

## 4.3 GRAPH LOTTERY TICKETS IN NODE CLASSIFICATION

Since ICPG can achieve excellent performance on diverse types and scales of graphs, we also want to explore that if it can also tackle node-level tasks. To answer this question, we conduct experiments on Core and PPI datasets, which are commonly used in transductive and inductive node classification tasks. We also reproduce the recent work UGS (Chen et al., 2021) for Cora (cannot apply for inductive setting) for better comparison. We can get the following **Obs**ervations:

**Obs.5. ICPG can achieve excellent performance on node classification tasks.** Firstly, for Cora, both ICPG and UGS can find GLTs that are sparser than RP, which demonstrates that both are applicable to transductive node classification. Secondly, ICPG can find sparser GLTs than UGS (↑7.94%), while the performance drop faster than UGS in the later stage, we give the following conjectures: (1) UGS just adopts simple trainable masks for edges without considering the global topological structure

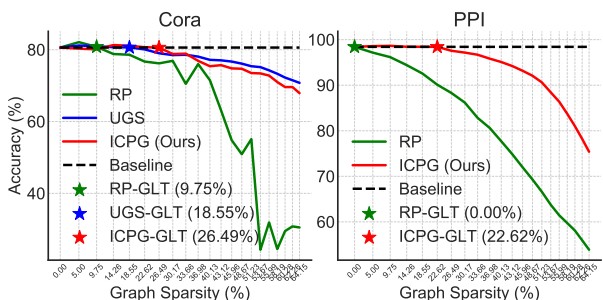

Figure 4: Transductive and inductive node classification performance on achieved graph sparsity.

of the entire graph, while AutoMasker is constructed based on a GNN-encoder, which can provide a global understanding of the whole graph. Hence, the AutoMasker can predict more high-quality masks than UGS. (2) ICPG is worse than UGS in the later stage. The reason is that AutoMasker has found the most significant edges in advance, which consists of an extremely compact core-subgraph at a certain critical point, so further pruning over that point will seriously degrade the performance. Thirdly, as for the PPI dataset, the performance of ICPG still keeps a large gap with RP and can achieve 22.62% graph sparsity and 67.23% GNN sparsity without sacrificing performance, which further demonstrates the effectiveness of the ICPG on inductive learning.

## 4.4 THE TRANSFERABILITY OF THE AUTOMASKER

We consider two orthogonal perspectives to verify the transferability of the AutoMasker: GNN-level transferability and graph-level transferability. From GNNs view, we transfer the sparse graphs found by AutoMasker to the other two popular GNN models: GIN (Xu et al., 2019) and GAT (Veličković et al., 2017). From graphs view, we first pre-train the AutoMasker on larger-scale RED-M12K graphs and then transfer the well-trained AutoMasker to other two smaller-scale graphs: RED-B and RED-M5K. Please notes that we keep the GNN models unpruned on transferred tasks. The experimental results are provided in Figure 5 and Table 1. We make the following **Obs**ervations:

**Obs.6. AutoMasker has both GNN-level and graph-level transferability.** For GNN-level, we can observe from Figure 5 that GIN and GAT can achieve ranging 9.75%∼45.96% and 18.55%∼22.62% extreme sparsity on NCI1 and RED-M12K without sacrificing performance. And AutoMasker also outperforms RP and keeps a large gap. These results demonstrate that AutoMasker can effectively extract the GNN-independent subgraphs. These sparse subgraphs represent significant semantic information and can be universally transferred to any GNN architecture without performance degradation. For graph-level, the performance of random pruning decreases as the graph sparsity increases. For RED-B and RED-M5K, when the pruning sparsity increases from 0 to 55.99%, the performance

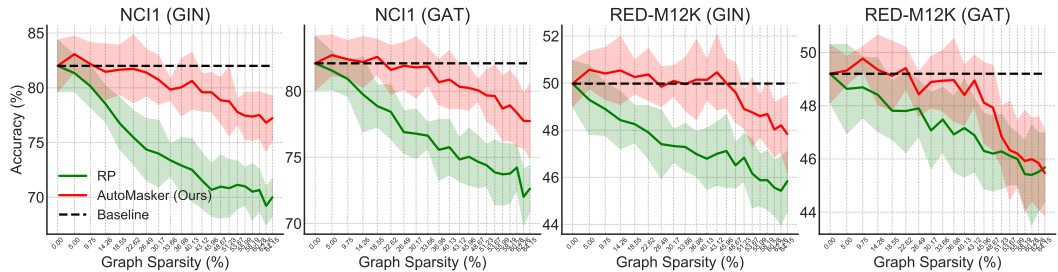

Figure 5: Performance of diverse GNNs on the sparse graphs found by AutoMasker.

Table 1: Graph-level transferability performance. From RED-M12K to RED-B and RED-M5K

| Settings | | Graph Sparsity | | | | | |
|---|---|---|---|---|---|---|---|
| Dataset | Method | 0% (No pruning) | 9.75 % | 18.55 % | 33.66 % | 45.96 % | 55.99 % |
| RED-B | Random Pruning | 92.15±1.59 | 90.60±1.22 | 89.75±1.75 | 86.75±2.41 | 85.15±3.92 | 85.34±1.67 |
| | AutoMasker | 92.15±1.59 | **92.16±2.06** | **91.05±2.14** | **90.15±1.89** | **90.06±2.57** | **89.64±1.72** |
| RED-M5K | Random Pruning | 56.63±0.93 | 56.33±1.59 | 55.85±1.08 | 54.81±2.17 | 54.19±2.27 | 54.95±1.79 |
| | AutoMasker | 56.63±0.93 | **56.89±2.16** | **56.69±2.59** | **57.01±3.90** | **56.97±2.86** | **56.09±4.44** |

decreases by 7.39% and 2.97%, respectively. While AutoMasker can achieve consistent improvement within all sparsity levels. Furthermore, the GNN model trained with more sparse graphs even outperforms the GNN trained with the original dense graphs, such as RED-B at 9.75% and RED-M5K at 9.75%∼45.96%. It demonstrates that the pre-trained AutoMasker can transfer the learned knowledge to small-scale downstream tasks, with lower computational cost and better performance. In summary, AutoMasker can learn model-independent, general and significant sparse subgraph structures from the graphs, so that it has outstanding GNN-level and graph-level transferability.

## 4.5 ABLATION STUDY

In this section, we investigate the diverse encoder networks in AutoMasker and explore the two independent components in ICPG. We can make the following findings:

**Encoder networks.** AutoMasker is designed on a GNN-based encoder, which leads to a global understanding of each edge from the entire graphs. So we extensively investigate the impact of the diverse encoders, such as GNN-based or MLP-based encoders. We can observe the results from Fig-

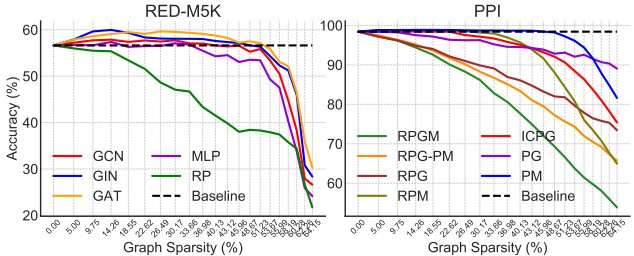

Figure 6: (**Left**): The performance of ICPG over achieved graph sparsity with AutoMasker based on diverse encoders. (**Right**): The comparison of different components in ICPG.

ure 6 (**Left**) that, for all the GNN-based encoders, AutoMasker can achieve good performance: 45.96% extreme sparsity for GCN and 51.23% for GIN and GAT, while MLP-based encoder only achieves 33.66% extreme sparsity. It indicates that the message passing scheme of the GNN encoder naturally considers the graph structure from a global view, while the MLP-based encoder does not.

**Co-sparsification.** To further study the effectiveness of each component in ICPG: mask-based pruning for graphs (PG) and magnitude-based pruning for model (PM), we separate them and explore their roles when applying on the graphs and the model independently. We also plot the performance of random pruning only for graphs (RPG), only for models (RPM), both of all (RPGM), random pruning for graphs with magnitude-based pruning for model (RPG-PM) for comparison. The results are summarized in Figure 6 (**Right**). We can find that: PG can also find the matching subgraphs (9.75% sparsity), which indicates that apply AutoMasker separately can also extract the significant edges from each graph. PM can also locate the matching subnetworks at 14.26% sparsity, which is consistent with the LTH (Frankle & Carbin, 2018) in the computer vision field. ICPG significantly outperforms RPGM and RPG-PM, and the gap gradually widens as the sparsity increases. We also observe that ICPG is even better than PG (↑12.87%), we made the following conjectures: (1) As for

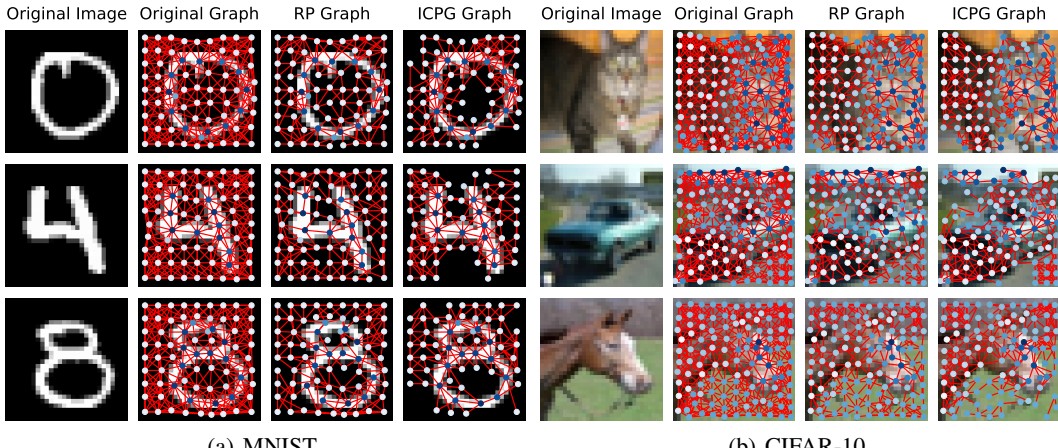

Figure 7: Visualization of the subgraphs extracted by AutoMasker from MNIST (a) and CIFAR-10 (b) superpixel graphs. Original images and graphs are displayed on the first and second columns in (a) and (b), respectively. The sparsity of RP and ICPG in (a) and (b) is 64.15%.

PG, with the sparsity gradually increasing, the graphs also become more simple. If we still train the over-parameterized GNN model with simple graphs, it may cause over-fitting. (2) Slightly pruning the over-parameterized GNN through PM can be regarded as a kind of regularization, which will improve the performance, and it is consistent with LTH (Frankle & Carbin, 2018). Further, the regularized GNN can additionally provide AutoMasker with more precise supervision signals from the gradient in backpropagation to make more wise decisions. To summary, we should co-train the AutoMasker and GNN and co-sparsify the input graphs and model to achieve better performance.

## 4.6 VISUALIZATION AND ANALYSIS

To better illustrate the significant subgraphs extracted by the AutoMasker, we visualize the matching subgraphs in GLTs found by the ICPG. We select graphs with 64.15% sparsity level from the MNIST and CIFAR-10 superpixel datasets. For better comparison, we also plot the original images, original graphs, random pruning graphs, which are depicted in Figure 7. More wonderful visualizations are provided in Figure A11 and Figure A12. We can make the following findings:

For MNIST and CIFAR-10, the edges between nodes that locate on the digitals and objects pixels (the dark blue nodes) should be denser, which are conducive to the graph classification tasks. RP evenly prunes the significant edges or structures without considering any important reference, which makes the core-subgraphs destroyed and seriously deteriorates the performance. ICPG utilizes AutoMasker to learn the significance of each edge from a global view and can precisely prune redundant edges. As the ICPG graph in Figure 7 (a) shows, the pruned edges are mainly located on non-digital pixels, such as the upper-left, lower-right corners and the center part of the number 0; the lower-left corner of the number 8, while the remaining edges or nodes are mainly located on digital pixels, which demonstrate that AutoMasker can indeed extract significant patterns.

## 5 CONCLUSION

In this work, we endow the graph lottery tickets with inductive pruning capacity. We propose a simple and effective pruning framework ICPG, to co-sparsify the input graphs and GNN model. For graphs, we propose a generative probabilistic model to generate the importance score for each edge. It provides a global understanding of edge importance from the entire graph topological structure to guarantee high-quality graph masks and has strong generalization ability and transferability in inductive learning settings. For the model, we adopt the GNN weight's magnitude to estimate their importance scores. Then we co-sparsify the input graphs and GNN model based on their important scores to find the graph lottery tickets. Extensive experiments on diverse types (biochemical molecules, social networks, superpixel graphs, citation networks) and scales (small, medium, large) of graphs, diverse learning settings (inductive, transductive) and diverse tasks (graph classification, node classification) consistently validate the effectiveness of the ICPG.

## 6 ETHICS STATEMENT

In recent years, graph neural networks (GNNs) have been widely applied in learning and processing graph-structured data, such as recommendation systems and drug discovery. With the rapidly growing size of graphs, deep graph neural networks will inevitably confront expensive computational costs and slow response times in the training and inference stage. Hence, it is challenging to implement deep GNNs in real-world large-scale graphs. However, the current GNNs pruning methods cannot solve the inductive learning settings, which are common in real-world scenes, such as dynamic social networks learning. Fortunately, the proposed co-pruning framework provides an effective and practical solution to reduce computational costs, which can be universally applied to both transductive and inductive graph learning. Furthermore, the proposed AutoMasker also gives a novel pre-training perspective to save the computational cost in downstream tasks.

## 7 REPRODUCIBILITY STATEMENT

We promise that all results in this paper are reproducible. The models, datasets, and training and inference settings in this paper have been given in the main text and Appendix A1. To help readers easily reproduce our results, we provide a more detailed explanation. Codes available at https://anonymous.4open.science/r/Inductive_Lottery_Ticket_Learning-1419

(1) **Models**. For simplicity, we keep the same GNN architecture for the proposed AutoMasker and the following GNN model. Please note that AutoMasker usually requires fewer hidden units. The GNN encoder network of AutoMasker can also be changed as needed and we have already given an example of replacing GCN based encoder with GIN or GAT in the ablation study. All the results in this paper are based on the GCN model, except for the PPI dataset. According to the suggestion of (Veličković et al., 2017), GAT (Veličković et al., 2017) network can achieve a better baseline performance on PPI dataset. More detailed model settings such as the number of layers or hidden units have been summarized in Appendix A1.4 and Table A3.

(2) **Datasets**. We conduct all the experiments on common graph learning benchmarks (Dwivedi et al., 2020; Xu et al., 2019; Veličković et al., 2017; Kipf & Welling, 2016; You et al., 2020; Hu et al., 2020). We have summarized the detailed information of the datasets in Appendix A1.2, the statistics of the datasets in Table A2, the method of the datasets splitting in Appendix A1.3.

(3) **Baselines**. We adopt the following two baselines for comparison: random pruning (RP) and UGS (Chen et al., 2021). For random pruning, we simply adopt a completely random selection strategy to prune the edges of the graph or the parameters of the GNN model. For UGS, we reproduce the results of the original paper (Chen et al., 2021) based on the released code.

(4) **Training and inference settings**. We extensively refer to the training settings for numerous literature in the field of graph learning (Chen et al., 2019; Xu et al., 2019; Dwivedi et al., 2020; Chen et al., 2021; Veličković et al., 2017), and configure the training hyperparameters according to their suggestions. The detailed training settings have been given in the Appendix A1.5. As for inference settings, we adopt the method consistent with (Chen et al., 2021; Xu et al., 2019; Hu et al., 2020). For the TUDataset, we adopt the 10-fold cross validation and report the mean and standard deviation. For the OGB datasets, we follow the official method (Hu et al., 2020) to report the results. For other datasets, we report the test accuracy at the epoch with the best validation accuracy.

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

## A1    MORE IMPLEMENTATION DETAILS

### A1.1    ALGORITHMS

We summarize the detailed implementation of the proposed ICPG in Algorithm 2. The Algorithm 1 represents the co-training and co-sparsifying for a single iteration in ICPG.

---

**Algorithm 1:** Mask & Magnitude Pruning

**Input:** $\mathcal{D}$, $f(\cdot, \boldsymbol{\Theta}_{g_0})$, AutoMasker($\cdot, \boldsymbol{\Theta}_{a_0}$), $\mathcal{M}$, $\mathbf{m}_\Theta$, Epoch $T$.
**Output:** Sparsified masks $\{\mathbf{m}'_{\mathcal{G}_i}\}_{i=1}^N$, $\mathbf{m}'_\Theta$.

1 **for** $t = 0$ *to* $T - 1$ **do**
2    **for** $\mathcal{G}_i \in \mathcal{D}$ *and* $\mathbf{m}_{\mathcal{G}_i} \in \mathcal{M}$ **do**
3      $\mathcal{G}_i \leftarrow (\mathbf{m}_{\mathcal{G}_i} \odot \mathbf{A}_i, \mathbf{X}_i)$
4      $\mathbf{s}_{\mathcal{G}_i} \leftarrow$ AutoMasker($\mathcal{G}_i, \boldsymbol{\Theta}_{a_t}$)
5      $\mathcal{G}_i \leftarrow (\mathbf{s}_{\mathcal{G}_i} \odot \mathbf{A}_i, \mathbf{X}_i)$
6      Forward $f(\mathcal{G}_i, \mathbf{m}_\Theta \odot \boldsymbol{\Theta}_{g_t})$
7      Backward to update $\boldsymbol{\Theta}_{a_{t+1}}, \boldsymbol{\Theta}_{g_{t+1}}$
8    **end**
9 **end**
10 **for** $\mathcal{G}_i \in \mathcal{D}$ **do**
11    $\mathbf{s}_{\mathcal{G}_i} \leftarrow$ AutoMasker($\mathcal{G}_i, \boldsymbol{\Theta}_{a_T}$)
12    Set $5\%$ of the lowest mask values in $\mathbf{s}_{\mathcal{G}_i}$ to 0 and others to 1, creating $\mathbf{m}'_{\mathcal{G}_i}$.
13 **end**
14 Prune $20\%$ of the lowest magnitude parameters in $\boldsymbol{\Theta}_{g_T}$, creating $\mathbf{m}'_\Theta$.

---

**Algorithm 2:** Finding GLTs by ICPG

**Input:** Graphs $\mathcal{D} = \{\mathcal{G}_i = (\mathbf{A}_i, \mathbf{X}_i)\}_{i=1}^N$, $f(\cdot, \boldsymbol{\Theta}_{g_0})$, AutoMasker($\cdot, \boldsymbol{\Theta}_{a_0}$), sparsity levels $s_d, s_\theta$.
**Output:** GLT $\{\mathcal{G}'_i = (\mathbf{m}_{\mathcal{G}_i} \odot \mathbf{A}_i, \mathbf{X}_i)\}_{i=1}^N$, $f(\cdot; \mathbf{m}_\Theta \odot \boldsymbol{\Theta}_{g_0})$.

1 Initialize masks set $\mathcal{M} \leftarrow \{\mathbf{m}_{\mathcal{G}_i} \leftarrow \mathbf{A}_i\}_{i=1}^N$
2 Initialize GNN mask $\mathbf{m}_\Theta \leftarrow \mathbf{1} \in \mathbb{R}^{\|\boldsymbol{\Theta}_{g_0}\|_0}$
3 **while** *the sparsity of* $\mathcal{M} < s_d$, $\mathbf{m}_\Theta < s_\theta$ **do**
4    Sparsify the GNN $f(\cdot; \boldsymbol{\Theta}_{g_0})$ with $\mathbf{m}_\Theta$ and dataset $\mathcal{D} = \{\mathcal{G}_i = (\mathbf{A}_i, \mathbf{X}_i)\}_{i=1}^N$ with the mask set $\mathcal{M}$ and get the new masks as presented in Algorithm 1.
5    Update $\mathcal{M} \leftarrow \{\mathbf{m}_{\mathcal{G}_i} \leftarrow \mathbf{m}'_{\mathcal{G}_i}\}_{i=1}^N$
6    Update $\mathbf{m}_\Theta \leftarrow \mathbf{m}'_\Theta$
7    Rewind AutoMasker's weight to $\boldsymbol{\Theta}_{a_0}$.
8    Rewind GNN's weight to $\boldsymbol{\Theta}_{g_0}$.
9 **end**

---

### A1.2    DATASETS DETAILS

As for graph classification, we conduct the experiments with three scale levels: small-scale, medium-scale and large-scale. We provide the following details:

- **Small-scale**: We adpot the TUDataset (Morris et al., 2020), which is a collection of benchmark datasets for graph classification and regression. We choose two biological graphs: NCI1, MU-TAG and four social graphs: COLLAB, Reddit-Binary (RED-B), Reddit-Multi-5K (RED-M5K), Reddit-Multi-12K (RED-M12K) as numerous works (Xu et al., 2019; You et al., 2020; Dwivedi et al., 2020) does.

- **Medium-scale**: We use MNIST and CIFAR-10 superpixel graphs. The original MNIST and CIFAR-10 images are converted to graphs using superpixels, which represent small regions of homogeneous intensity in images and can be extracted with the SLIC (Achanta et al., 2012) technique. These datasets are commonly used in numerous graph representation learning researches (Dwivedi et al., 2020; You et al., 2020; Knyazev et al., 2019).

- **Large-scale**, we adopt the Open Graph Benchmark (OGB) (Hu et al., 2020), which is a collection of realistic, large-scale, and diverse benchmark datasets for machine learning on graphs. We choose two commonly used large-scale graph property prediction dataset: ogbg-ppa and ogbg-code2. The ogbg-ppa dataset is a set of undirected protein association neighborhoods extracted from the protein-protein association networks of 1,581 different species that cover 37 broad taxonomic groups and span the tree of life. The ogbg-code2 dataset is a collection of Abstract Syntax Trees (ASTs) obtained from approximately 450 thousands Python method definitions.

As for node classification, we use the commonly used citation network Cora dataset for semi-supervised learning and the popular inductive learning dataset PPI. The task for PPT is classifying protein functions across various biological protein-protein interaction (PPI) graphs. These datasets are commonly used in transductive learning (Kipf & Welling, 2016; Veličković et al., 2017) and inductive learning (Hamilton et al., 2017; Zheng et al., 2020) researches. All the detail statistics about theaforementioned datasets are summarized in Table A2.

Table A2: Datasets statistics.

| Datasets | Category | Graphs | Avg. Nodes | Avg. Edges | Avg. Degree | Classes |
|---|---|---|---|---|---|---|
| NCI1 | Biochemical Molecules | 4,110 | 29.87 | 32.30 | 1.08 | 2 |
| MUTAG | Biochemical Molecules | 188 | 17.93 | 19.79 | 1.10 | 2 |
| COLLAB | Social Networks | 5,000 | 74.49 | 2457.78 | 32.99 | 3 |
| RED-B | Social Networks | 2,000 | 429.63 | 494.07 | 1.15 | 2 |
| RED-M5K | Social Networks | 4,999 | 508.52 | 594.87 | 1.17 | 5 |
| RED-M12K | Social Networks | 11,929 | 391.41 | 456.89 | 1.16 | 11 |
| MNIST | Superpixel Graphs | 70,000 | 70.57 | 564.56 | 8.00 | 10 |
| CIFAR-10 | Superpixel Graphs | 60,000 | 117.63 | 941.04 | 8.00 | 10 |
| ogbg-ppa | OGB Dataset | 158,100 | 243.4 | 2,266.1 | 9.31 | 37 |
| ogbg-code2 | OGB Dataset | 452,741 | 125.2 | 124.2 | 0.99 | - |
| Cora | Citation Network | 1 | 2708 | 5429 | 2.00 | 7 |
| PPI | Biological Protein | 24 | 2372.67 | 34113.16 | 14.38 | 121 |

## A1.3 DATASETS SPLITTING

**TUDataset**: We perform the commonly used 10-fold cross validation. Consistent with the work (Chen et al., 2019; Xu et al., 2019), we select the epoch with the best cross-validation accuracy averaged over the 10 folds and report the average and standard deviation of test accuracies at the selected epoch over 10 folds.

**Superpixel dataset**: Consistent with (Dwivedi et al., 2020), we split them to 55000 train/5000 validation/10000 test for MNIST, and 45000 train/5000 validation/10000 test for CIFAR10, respectively. We report the test accuracy at the epoch with the best validation accuracy.

**OGB dataset**: We adopt the official dataset splitting method for ogbg-ppa and ogbg-code2 in the following links: `https://ogb.stanford.edu/docs/graphprop/#ogbg-ppa` and `https://ogb.stanford.edu/docs/graphprop/#ogbg-code2`.

**Cora**: Following the work (Chen et al., 2021), we use 140 labeled data for training, 500 nodes for validation and 1000 nodes for testing and report the test accuracy at the epoch with the best validation accuracy.

**PPI**: We adopt the same splitting as works (Hamilton et al., 2017; Veličković et al., 2017). The dataset contains 20 graphs for training, 2 for validation and 2 for testing. Critically, testing graphs remain completely unobserved during training.

## A1.4 MODEL CONFIGURATIONS

TUDataset: we adopt the ResGCN (Chen et al., 2019) with 3 layers and 128 hidden dimensions. Superpixel dataset: we use GCN with 4 layers and 146 hidden dimensions as work (Dwivedi et al., 2020). OGB dataset: we use five-layer GCN model with 300 hidden dimensions for all experiments. Cora dataset: we adopt two-layer GCN network with 512 hidden dimensions as work (Chen et al., 2021). PPI dataset: we adopt two-layer GAT network with 256 hidden and 4 attention heads as work (Veličković et al., 2017). For GCN encoder in AutoMasker, we use the same structure with different hidden dimensions as the GNN model. More details are summarized in Table A3.

## A1.5 TRAINING DETAILS

All training hyper-parameters such as epochs, learning rate for GNN model and AutoMasker, optimizer, batch size, weight decay are summarized in Table A3.

Table A3: Implementation details of graph classification and node classification.

| Task | Graph Classification | | | | Node Classification | |
|---|---|---|---|---|---|---|
| Dataset | TUDataset | Superpixel | ogbg-ppa | ogbg-code2 | Cora | PPI |
| Epoch | 100 | 100 | 100 | 25 | 200 | 100 |
| Optimizer | Admm | Admm | Admm | Admm | Admm | Admm |
| Batch Size | 128 | 128 | 32 | 128 | 1 | 1 |
| Weight Decay | 0 | 0 | 0 | 0 | 5e-4 | 0 |
| Model Layer-hidden | 3-128 | 4-146 | 5-300 | 5-300 | 2-512 | 2-256 |
| AutoMasker Layer-hidden | 3-64 | 4-146 | 5-300 | 5-300 | 2-128 | 2-128 |
| Model LR | 1e-3 | 1e-3 | 1e-3 | 1e-3 | 1e-2 | 5e-3 |
| AutoMasker LR | 1e-4 | 5e-3 | 1e-3 | 1e-3 | 1e-2 | 1e-3 |

## A2 MORE EXPERIMENTAL RESULTS

### A2.1 GRAPH LOTTERY TICKETS IN GRAPH CLASSIFICATION

The graph classification performance with different GNN sparsity levels is shown in Figure A8. The performance on large-scale datasets is provided in Figure A9.

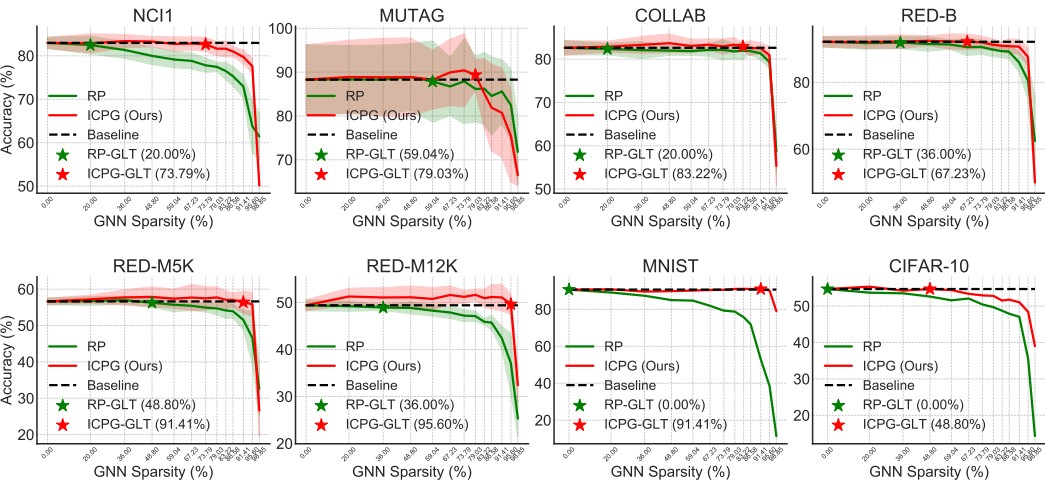

Figure A8: Graph classification performance over achieved GNN sparsity.

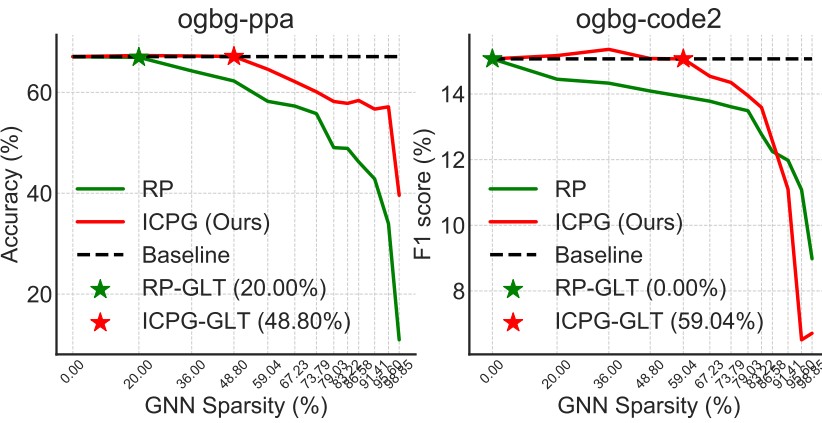

Figure A9: Graph classification performance over achieved GNN sparsity on large-scale datasets.

## A2.2   GRAPH LOTTERY TICKETS IN NODE CLASSIFICATION

More results about the transductive and inductive node classification on achieved GNN sparsity are shown in Figure A10. We can observe that ICPG can achieve 73.79% GNN sparsity for transductive node classification and 67.23% GNN sparsity for inductive node classification.

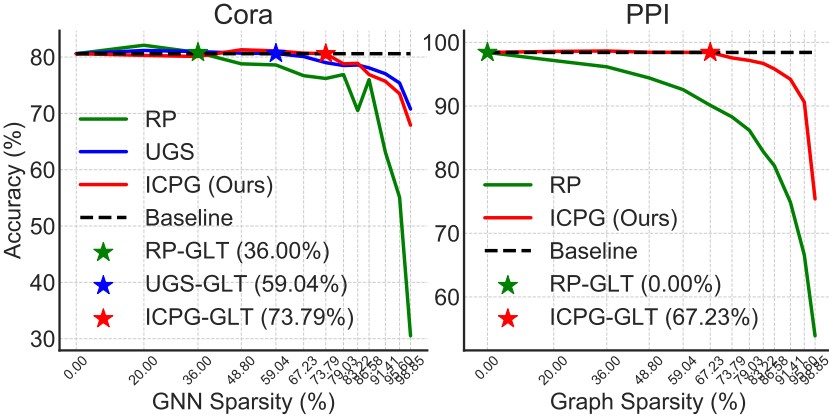

Figure A10: Transductive and inductive node classification performance on achieved GNN sparsity.

## A3   MORE VISUALIZATION RESULTS

More visualization results about the MNIST and CIFAR-10 superpixel graphs are shown in Figure A11 and Figure A12. For MNIST, the important subgraphs are mainly located on the digital pixel area. We can observe that RP evenly prunes the connections between any two nodes without considering any important information. While the proposed AutoMasker mainly prunes the insignificant edges that locate on non-digital pixel background. For CIFAR-10, objects are critical information for classification, so the edges between the nodes that locate on the objects (the dark blue nodes) should be denser. We can observe that RP prunes on both the objects and the background, which also makes all the vital edges become sparse. As a comparison, the subgraphs extracted by AutoMasker remain most of the edges located on objects while the sparsify edges are located on the backgrounds. In summary, the AutoMasker effectively extracts the significant core patterns while filter out the redundant edges from dense graphs, which necessarily leads to less degradation on performance even the graphs are heavily pruned.

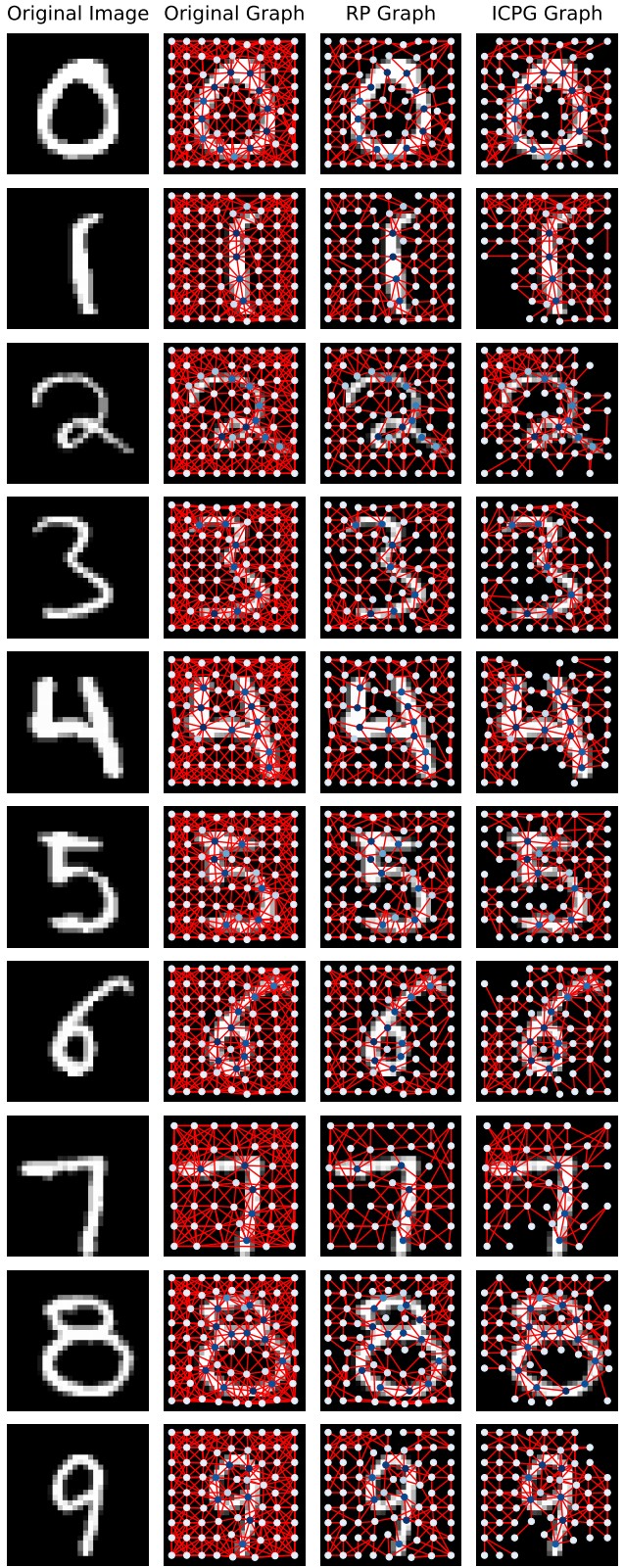

Figure A11: Visualization of the subgraphs at $64.15\%$ sparsity from MNIST superpixel graphs.

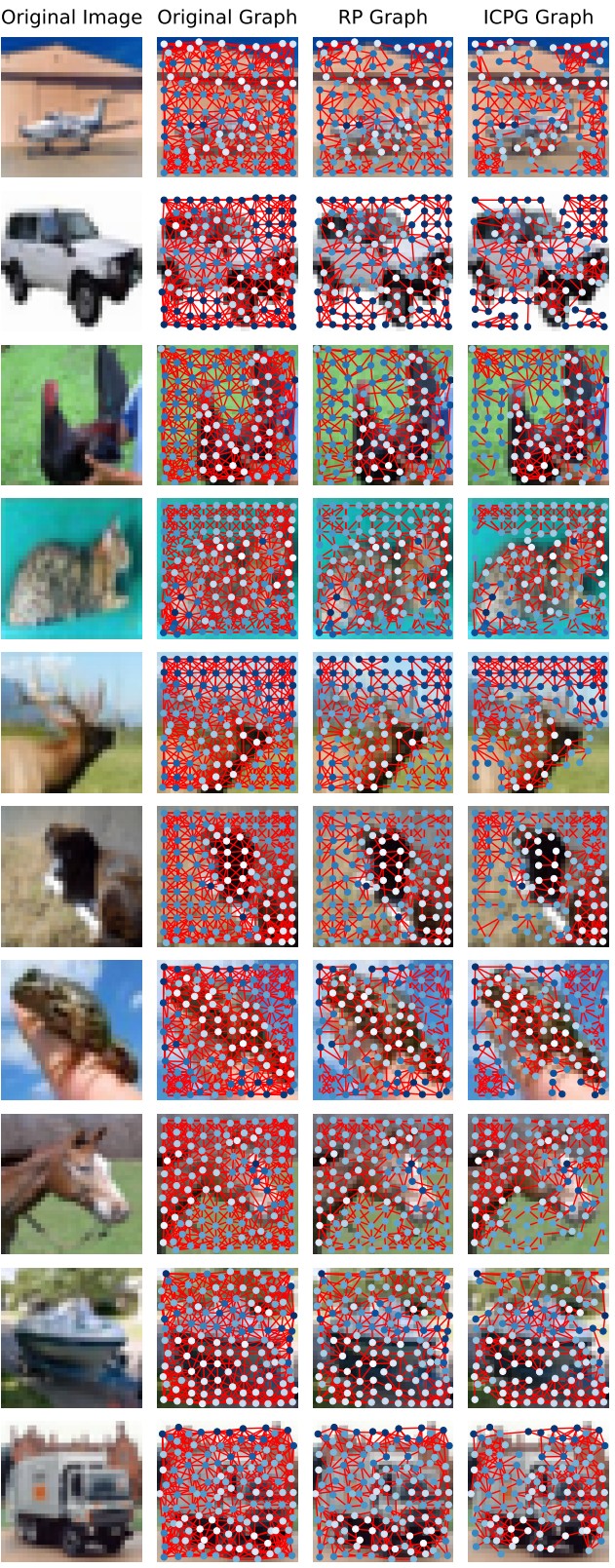

Figure A12: Visualization of the subgraphs at $64.15\%$ sparsity from CIFAR-10 superpixel graphs.

