# OpenReview forum: "Inductive Lottery Ticket Learning for Graph Neural Networks"
_ICLR.cc/2022/Conference — ICLR 2022 Submitted_

### Official Review · Reviewer_4F4b · 2021-11-02

**Correctness:** 3
**Technical Novelty And Significance:** 2
**Empirical Novelty And Significance:** 3
**Recommendation:** 6
**Confidence:** 3

**Main Review:**

It is very important to extend the concept of methodologies for the transductive setting to that of the inductive setting in GNNs.

This work, ICPG, has significant meaning on this point but it is also true that it seems like the simple extended (or improved) version of UGS.

Explicitly, the most novel component of this framework is the AutoMasker (and also co-training strategy is a key idea as well, but there are a lot of co-training frameworks for GNNs).

AutoMasker has a simple architecture (GNN-MLP) and provides the importance scores of edges like EdgePool [1].

Even though the authors described several conceptual advantages over UGS, I wonder if there is a better design than this (do you have several candidates for it?).

[1] Edge contraction pooling for graph neural networks, F Diehl

**Summary Of The Paper:**

In this work, the authors propose an iterative co-pruning framework using a lottery ticket learning for GNNs.

Graph lottery ticket learning implies that a sparse subnetwork exists in a dense and randomly initialized network for both GNNs and graph data.

UGS was proposed by Chen et al. to apply the lottery ticket learning to GNNs.

The basic concept is quite similar but it is distinguished from Chen et al. in that it can be applied for the inductive setting.

In Inductive Co-Pruning of GNNs (ICPG), edges of input graphs and model parameters are pruned according to the importance scores for each.

The authors exploit generative probabilistic models named AutoMasker to specify the importance of each edge and then configure core subgraphs.

Compared to the previous Graph Lottery Ticket (GLT) model, ICPG has a nice generalization ability for both node classification and graph classification.

**Summary Of The Review:**

I couldn't find an issue to reject this paper but it is difficult to evaluate novelty enough.
To be honest, it is quite simple so there is not much to comment on but at the same time, the contribution is clear.
I want to make a decision after discussing it with other reviewers or reading other points that I might miss.

---

> ### Author Response · Authors · 2021-11-13
> **Response to Reviewer 4F4b [Con 1]**
>
> Thank you for your careful reading and positive ratings. To address your concerns, we give point-to-point responses:
>
> **[Con 1: Simple extended version of UGS and limited novelty]** I respectfully disagree with this point. Here we give the following reasons:
>
> -	**For UGS contribution:** UGS [1] for the first time to find the lottery tickets in graph learning, but it does not propose a good pruning strategy. The UGS framework is very simple and intuitive. It has many limitations, which we have discussed in the introduction. These limitations are fatal and make graph lottery tickets unable to extend to inductive learning, which is very important and common for the graph learning field as you said.
> -	**For method:** Our method and UGS are very different. For graph-level, we adopt AutoMasker to evaluate the importance of each edge from a global perspective. It flexibly makes our framework suitable for both transductive and inductive learning settings. While UGS just uses trainable weights as an indicator to prune the graph, which is local and suboptimal. For GNN-level, UGS learns a trainable mask for each model parameter. It doubles the computational cost in the training stage.  Different from them, we use more lightweight magnitude-based pruning, which greatly reduces the computational cost.
> -	**For our contribution:** To our best knowledge, we are the first to achieve data and model co-pruning for inductive graph learning. We successfully extend lottery tickets to a wider range of graph learning tasks. Specifically, the proposed framework can flexibly develop on graphs with different types (molecules, superpixel graphs, social networks, citation networks), different scales (from small-scale to large-scale), diverse learning settings (inductive, transductive), and various tasks (node, graph classification). It proves that the proposed framework is more general, universal and applicable to many fields in graph learning.  In other words, we bring new ideas and open a new direction for researchers to study pruning on more graph learning applications.  In addition, the proposed AutoMasker has excellent transferability. On one hand, GNN-level transferability points out that the remaining subgraphs contain significant semantic information, which gives more insights for graph explainability [2-4]. The wonderful visualizations provided in Figure 7 also give us strong confidence. On the other hand, graph-level transferability opens a new direction for GNN pre-training [5]: we can make full use of the general knowledge learned from upstream tasks and transfer the knowledge to downstream tasks for efficient GNN training. It achieves the double-win: saving large computational costs and achieving better performance.
>
> [1] A unified lottery ticket hypothesis for graph neural networks, ICML'2021
>
> [2] Explainability in graph neural networks: A taxonomic survey.
>
> [3] Generative causal explanations for graph neural networks, ICML'2021
>
> [4] Gnnexplainer: Generating explanations for graph neural networks, NeurIPS'2019
>
> [5] Graph contrastive learning with augmentations, NeurIPS'2020

---

> ### Author Response · Authors · 2021-11-13
> **(Continued) Response to Reviewer 4F4b [Con 2] and [Con 3]**
>
> **[Con 2: Co-training framework is common for GNNs]**
> Existing co-training frameworks [6-10] for GNNs mainly focus on how to learn a robust graph structure.  These works are very different from ours. According to your concerns, we give the following points:
>
> -	**The purpose is different.** These works [6-10] make efforts to find a better or robust graph structure, thereby improving generalization or robustness. While our proposal aims at co-pruning the data and model. We hope we can capture more compact graph structures and lightweight GNN models to save computational costs without sacrificing performance.
> -	**Existing works are difficult to develop on inductive learning.** They mainly focus on a single graph with transductive learning settings, such as node classification on the Cora dataset. Various optimization methods, such as ADMM [11], can be easily applied to a given graph. However, in inductive learning settings, there exist numerous graphs with different nodes, which leads to different sizes of adjacency matrices. It is difficult for the current methods to deal with these irregular matrices. To overcome the dilemma, we use the AutoMasker to universally project and align diverse graphs into the same latent space for optimization, thereby avoiding directly optimizing various adjacency matrices with irregular sizes.
> -	**Providing new ideas to graph structure learning.** Although our purpose is pruning graphs, we also provide a new idea for graph structure learning.  We can further explore more the high-quality graph structures based on the AutoMasker. It opens a new direction for researchers that we can also extend graph structure learning to more graph learning applications with inductive learning settings.
>
> [6] Iterative Deep Graph Learning for Graph Neural Networks: Better and Robust Node Embeddings, NeurIPS'2020
>
> [7] Deep Iterative and Adaptive Learning for Graph Neural Networks, AAAI'2020
>
> [8] Learning Discrete Structures for Graph Neural Networks, ICML'2019
>
> [9] Graph agreement models for semi-supervised learning, NeurIPS'2019
>
> [10] Graph Structure Learning for Robust Graph Neural Networks, KDD'2020
>
> [11] Sgcn: A graph sparsifier based on graph convolutional networks, PAKDD'2020
>
>
> **[Con 3: EdgePool]** Although EdgePool [12] also adopts the importance score to evaluate each edge, our work is quite different from them. We list the following points:
>
> -	**Different purposes.**  EdgePool makes efforts to improve the graph pooling layer, thereby improving the generalization performance of the model. Our purpose is to find sparse graphs for GNNs, so as to ensure performance while saving computational costs.
> -	**Different score calculation methods.** EdgePool uses a local function to calculate the score of the edges connected to a given central node. It locally computes the scores among the neighbors. However, we adopt a GNN-based auxiliary model to globally predict scores for all edges. It evaluates the importance of each edge from a global perspective and achieves more precise pruning.
> -	**Different ways to extract subgraphs.** EdgePool uses the edge with the highest score to merge nodes, thereby reducing the number of nodes. Different from them, we prune the edges based on the lowest score, so as to capture subgraphs with sparser edges.
>
> Although there are many differences, we are happy to compare the EdgePool with our methods (51.23% graph sparsity-level). The results are shown in Table S1. We find that our method outperforms EdgePool in most cases, which further demonstrates the superiority of our method.
>
>
>
> Table S1. Test Accuracy (%) of graph classification. We perform 10-fold cross-validation to evaluate the performance, and report the mean and standard derivations.  Improvements denote the improved accuracy.
>
> | Methods      |      NCI1      |    PROTEINS    |     COLLAB     |     RDT-B      |     RDT-5K     |
> | :----------- | :------------: | :------------: | :------------: | :------------: | :------------: |
> | EdgePool     | 80.15$\pm$1.35 | 72.88$\pm$1.20 | 68.32$\pm$1.73 | 87.28$\pm$3.10 | 52.88$\pm$2.26 |
> | Our Method   | 77.62$\pm$1.75 | 74.14$\pm$4.97 | 80.92$\pm$1.98 | 87.65$\pm$2.79 | 55.85$\pm$3.31 |
> | Improvements |     ↓2.53      |     ↑1.26      |     ↑12.6      |     ↑0.37      |     ↑2.97      |
>
> [12] Edge contraction pooling for graph neural network.

---

> ### Author Response · Authors · 2021-11-13
> **(Continued) Response to Reviewer 4F4b [Con 4]**
>
> **[Con 4: Better design candidates]**
> For graph data, we use a GNN-based predictive model to achieve inductive pruning.  For the model, we adopt magnitude-based pruning to achieve a lightweight GNN. ICPG uses multiple iterations to find GLTs.  Based on the above description, we list the following improvements as our future works.
>
> **(1) For framework improvements.**
>
> -	**Explore better GNN-based encoder.** The AutoMasker adopts a GNN-based backbone to make edge pruning. From the ablation study, we observe that different GNN architectures achieve different performances. So it is necessary to explore a better GNN backbone to make our AutoMasker stronger.
> -	**Explore better model pruning strategy.** In addition to magnitude-based pruning, there exist other ways to guide the model pruning, such as gradient-based pruning [13].  We can also  explore better model pruning methods to further improve the performance.
>
> **(2) For efficient GLTs finding.**
>
> -	**Explore the early stop mechanism.** Early-bird tickets [14] proposes an early stop strategy to locate the lottery tickets at an earlier stage. It greatly saves the computational cost to find the lottery tickets. This work also inspires us to explore the early stop mechanism in the co-training stage (step 1) to locate the GLTs efficiently.
> -	**Less data is more.** In our framework, we use all the training data to find the GLTs. Recent work [15] proposes a data-efficient ticket finding strategy to further save the computational cost. Considering graph classification tasks, we can also borrow similar ideas to just adopt the core-subset of the training graphs to locate the GLTs.
> -	**Dynamic sparse training.** The work [16] proposes a novel pruning framework to dynamically prune the network parameters in the training stage. It also greatly saves the computational cost and achieves similar performance as the IMP [17]. We can also explore this research line to further improve our work.
>
> [13] Pruning convolutional neural networks for resource efficient inference.
>
> [14] Drawing early-bird tickets: Towards more efficient training of deep networks, ICLR'2020
>
> [15] Efficient lottery ticket finding : Less data is more, ICML'2021
>
> [16] Rigging the lottery : Making all tickets winners, ICML'2020
>
> [17] The Lottery Ticket Hypothesis: Finding Sparse, Trainable Neural Networks, ICLR'2019

---

> ### Author Response · Authors · 2021-12-04
> **Looking foward to your reply!**
>
> Dear Reviewer 4F4b,
>
> Thank you for your reviews. We want to know if our response address your concerns.
> Please kindly let us know if there is anything else we can address to convince you for upgrading the scores.
>
> Best wishes,
>
> Authors

---

### Official Review · Reviewer_HhcH · 2021-11-04

**Correctness:** 3
**Technical Novelty And Significance:** 2
**Empirical Novelty And Significance:** 2
**Recommendation:** 5
**Confidence:** 3

**Main Review:**

The paper aims to solve an important real-life problem - that of pruning inductive graphs. The idea in the algorithm is pretty simple - 1) learn a mask for edges using the features of the nodes connected by the nodes, and 2) apply a gnn to the graph modified using the mask in step 1.

I did not find any insights that generalize from the paper. The proposed method is one way to perform pruning a graph and generating a GLT, but there could be several ways. The experiments also compare the method to random pruning alone, which is a weak baseline. For instance, it would be really interesting to see how the inductive properties of the graph help, which can be done by comparing the performance to the transductive pruning algorithm.

Here are some of my other questions and concerns.
- The authors use the term 'generative probabilitistic model' in the abstract to generate importance scores. Is this referring to the sigmoid and MLP in equation (4)? This is a discriminative model, and I do not see what is generative about it.
- The authors criticize UGS for independently pruning edges (in the experiments), and also for simultaneously pruning the graph and weights (in point 3 in the introduction). This seems contradictory. Pruning edges independently has the benefit of being robust to unsatisfactory edge masks (which is listed as the shortcoming of simultaneous pruning).
- One of the claimed advantages of the mask is that it takes the global graph into account and is not independent for every edge. Eqn 4 however only takes the incident node features into account. While this is global because the MLP is trained for all the edges, it does ignore the neighborhood of an edge in deciding whether to mask it or not. For example, if a graph has random / useless node features but useful neighborhood structure, the mask should utilize this information.
- What does it mean when s_G[i,j] is high for an edge that did not originally exist? It seems incorrect to ignore the high value s_G[i,j] for such edges - for example this may be telling us about a missing edge.
- The term extreme sparsity is not defined. This makes it difficult to follow the experiments section.
- Please compare to stronger baselines than random pruning. For instance, how do transductive pruning algorithms perform?

Some minor comments:
- The notation G' = (mG \hadamard A, X) is used in Sec 3.1. Wouldn't the features X be affected as well by mG?
- The last sentence in Obs.1 does not make sense.

**Summary Of The Paper:**

This paper proposes a method to perform pruning of graph neural networks. The novelty of the paper is that the method works for inductive graphs, and thus can generalize to unseen graphs. By virtue of the masks being inductive, they also have the potential to take the global graph into consideration and not apply locally on each edge.

**Summary Of The Review:**

Overall, I feel that the paper identifies an important problem, but the solution proposed and experiments are not thorough. The algorithm can be refined by incorporating neighborhood information in the mask generation process. The empirical performance can be vetted by comparing with existing pruning baselines.

---

> ### Author Response · Authors · 2021-11-13
> **Response to Reviewer HhcH [Con 1]**
>
> Thank you for your time. It seems that you have missed some important information in our paper. We prudently provide point-to-point responses to address your concerns. We believe you will change your attitude if you read them carefully.
>
> **[Con 1: Insights about the paper]** We respectfully argue our novelty is strong and we list the following points:
>
> -	**For method**: I think you may have overlooked some important information . Instead of simply using MLP and node features, the AutoMasker adopts an auxiliary GNN-based network as a backbone to learn node representation. Based on the message-passing mechanism of GNN, it incorporates the neighborhood information into the node representation. Then the MLP and sigmoid function generate the importance score for each edge. Benefiting from GNN, AutoMasker takes a global view of the graph population, which enables us to identify the edge coalitions. Specifically, edges usually collaborate with each other to make predictions, rather than working individually. Considering such coalition effects, AutoMasker is able to measure the importance of edges more accurately.
> -	**For contribution**: To our best knowledge, we are the first to achieve graph and model co-pruning for inductive graph learning. We successfully extend lottery tickets to a wider range of graph learning tasks. Specifically, the proposed framework can flexibly develop on graphs with different types (molecules, superpixel graphs, social networks, citation networks), different scales (from small-scale to large-scale), diverse learning settings (inductive, transductive), and various tasks (node, graph classification). It proves that the proposed framework is more general, universal and applicable to many fields in graph learning.  In other words, we bring new ideas and open a new direction for researchers to study pruning on more graph learning applications.
> -	**For other practical usages**:  In section 4.4, we conclude that AutoMasker has excellent transferability. On one hand, GNN-level transferability points out that the remaining subgraphs contain significant semantic information, which gives more insights for graph explainability [1-3]. The wonderful visualizations provided in Figure 7 also give us strong confidence. On the other hand, graph-level transferability opens a new direction for GNN pre-training [4]: we can adopt the general knowledge learned from upstream tasks and transfer the knowledge to downstream tasks for efficient GNN training. It achieves the double-win: saving large computational costs and achieving better performance.
> -	**For other comments:** Finally, our novelty and contributions were also thankfully appreciated by Reviewer 6qg1 and Reviewer 4F4b, commented as “novel”, “impressive” and “significant meaning”.  As Reviewer 4F4b said: “It is very important to extend the concept of methodologies for the transductive setting to that of the inductive setting in GNNs.” As Reviewer 6qg1 said: “many GNN based algorithms fail to generalize on large graphs.”
>
> [1] Explainability in graph neural networks: A taxonomic survey.
>
> [2] Generative causal explanations for graph neural networks, ICML'2021
>
> [3] Gnnexplainer: Generating explanations for graph neural networks, NeurIPS'2019
>
> [4] Graph contrastive learning with augmentations, NeurIPS'2020

---

> ### Author Response · Authors · 2021-11-13
> **(Continued) Response to Reviewer HhcH [Con 2]**
>
> **[Con 2: Stronger baselines]** First of all, we have compared with the state-of-the-art transductive pruning algorithm UGS [5] in section 4.3. Our main contribution is to achieve graph and GNN co-pruning for inductive graph learning. As Reviewer 6qg1 said: “the authors don't have a comparable baseline from previous works” (e.g. inductive pruning methods). Existing state-of-the-art graph pruning methods [5-7] mainly focus on a single graph under transductive settings. Unfortunately, they are difficult to develop in inductive learning settings, such as graph classification. To this end, our pruning framework is specially designed to overcome this dilemma (which is one of our main contribution). To argue the effectiveness, we adopt two techniques as stronger baselines: GraphSAGE [8] and DropEdge [9], which can easily apply to inductive graph classification tasks.  For a fair comparison, we adjust the hyper-parameters in GraphSAGE to achieve similar sparsity-levels with others. The results are shown in Table S1. We can find that our method consistently outperforms other baselines.  This further demonstrates the superiority of ICPG.
>
> Table S1. Test Accuracy (%) of graph classification across different sparsity-levels. We perform 10-fold cross-validation to evaluate the performance, and report the mean accuracy.  Improvements denote the improved accuracy compared with the best baseline.
>
> | Method       | NCI1 (14.26%, 22.62%, 33.66%, 40.13%) | COLLAB (14.26%, 22.62%, 33.66%, 40.13%) | RED-B (14.26%, 22.62%, 33.66%, 40.13%)  | RED-M5K (14.26%, 22.62%, 33.66%, 40.13%) |
> | :----------- | :-----------------------------------------------: | :-------------------------------------------------: | :------------------------------------------------: | :--------------------------------------------------: |
> | GraphSAGE    |           (77.79, 76.62, 72.97, 71.27)            |            (75.48, 73.93, 69.04, 68.50)             |            (84.32, 79.49, 79.35, 72.15)            |             (50.10, 47.65, 44.25, 36.83)             |
> | DropEdge     |           (83.09, 82.24, 81.40, 80.14)            |            (82.91, 82.16, 81.96, 81.52)             |            (91.40, 89.60, 88.00, 87.40)            |             (52.47, 50.37, 46.85, 45.35)             |
> | Our Method   |           (83.28, 82.82, 81.63, 80.34)            |            (83.68, 83.34, 82.90, 82.44)             |            (92.45, 92.15, 90.95, 90.10)            |             (57.87, 57.69, 57.07, 56.63)             |
> | Improvements |           (↑0.19, ↑0.58,  ↑0.23, ↑0.20)           |            (↑0.77, ↑1.18, ↑0.94, ↑0.92)             |            (↑1.05, ↑2.55, ↑2.95, ↑2.7)             |            (↑5.40, ↑7.32, ↑10.22, ↑11.28)            |
>
> [5] A unified lottery ticket hypothesis for graph neural networks, ICML'2021
>
> [6] Sgcn: A graph sparsifier based on graph convolutional networks, PAKDD'2020
>
> [7] GEBT: Drawing Early-Bird Tickets in Graph Convolutional Network Training
>
> [8] Inductive representation learning on large graphs, NeurIPS'2017
>
> [9] Dropedge: Towards deep graph convolutional networks on node classification, ICLR'2020

---

> ### Author Response · Authors · 2021-11-13
> **(Continued) Response to Reviewer HhcH [Con 3]**
>
> **[Con 3: Other questions and concerns]**
>
> **(1) For "generative probabilitistic model".** What we want to express is that the AutoMasker can generate the probability value of whether each edge exists. The "generative" here is not the meaning of a generative model. Considering that this expression may cause confusion, we adopt your valuable suggestions and change the term to "predictive model". Thanks for the suggestion.
>
> **(2) For UGS.** The "independently" that we criticize is that the UGS only provides a local (or independent) understanding about each edge. We did not argue that the co-pruning in UGS is not good. UGS assigns a trainable weight to each edge independently, and the weight indicates the importance of each edge. We criticize that this weight learning strategy only provides a local understanding for each edge, but does not fully consider the global view. What we are arguing is the poor graph pruning strategy in UGS. The suboptimal graph pruning strategy will seriously destroy the semantic information in original graphs. Since the GNN mask and the graph mask are trained and pruned simultaneously in UGS, the following trainable parameters trained on the wrong graph data will inevitably lead to suboptimal pruning for the GNN model. This will also feedback to the trainable edge mask for graph and cause worse graph pruning, thus forming a vicious circle. Thank you for your careful reading, we have revised the paper to make it more clear.
>
> **(3) For global view.** We think you have missed some important information about AutoMasker. In section 3.2, we introduced our proposed AutoMasker in detail.  First, it adopts an auxiliary GNN network as a backbone to learn node representation in Equation 3. The GNN-based network will incorporate the topological information of the entire graph into node representations. Next, the MLP and sigmoid function in Equation 4 project the concatenated node representations to the probability. Therefore, the key to achieving a global understanding is the GNN-based network. It incorporates the neighborhood information in the mask generation process.
>
> **(4) For $\mathbf{s}_{\mathcal{G}}[i,j]$.** I think you have proposed an interesting idea: we can generate a score to predict the existence of edge between two nodes that are not connected. It can generate useful new edges to improve the distinguishability of the graph. However, in this work, we mainly focus on how to prune the graph data and GNN model to reduce the large computational cost in training and inference, without sacrificing the performance. We do not care about the inexistent edges in this work. Even so, we will adopt your valuable and insightful proposal, and use AutoMasker to explore the edge-generating task in future work.
>
> **(5) Extreme sparsity.** We have already explained this term in the 4th sentence "stars denote the extreme sparsity" in section 4.2. It denotes a certain sparsity level.  After exceeding this level, the performance will be worse than the baseline (full model with full graphs). Since this term is commonly used in lottery ticket fields [10-12], for brevity, we did not give such a detailed explanation when writing this paper. Even so, we apologize that we did not make this clear and we have provided a clearer definition of these terms in the updated manuscript.
>
> **(6) Stronger baselines.** Please refer to **[Con 2: Stronger baselines]** above.
>
> [10] The lottery tickets hypothesis for supervised and self-supervised pre-training in computer vision models, CVPR'2021
>
> [11] The lottery ticket hypothesis for pre-trained bert networks, NeurIPS'2020
>
> [12] GANs can play lottery tickets too, ICLR'2021

---

> ### Author Response · Authors · 2021-11-13
> **(Continued) Response to Reviewer HhcH [Con 4] and [Con 5]**
>
> **[Con 4: Some minor comments]**
>
> (1) The adjacency matrix $\mathbf{A}$ reflects the connections between nodes, and the node feature matrix $\mathbf{X}$ denotes the properties of the nodes themselves.  They are independent, so pruning edges (applying $\mathbf{m}_{\mathcal{G}}$) will not affect the node feature.
>
> (2) Thank you for careful reading and we have revised the Obs. 1 in the draft.
>
>
>
> **[Con 5: Summary of the review]**
>
> **(1) For the experiments**. We respectfully disagree with your comments "not thorough".  We conduct extensive experiments on 12 datasets, which include:
>
> -	Different types of graph: biochemical molecules, superpixel graphs, social networks, citation networks, which are commonly used benchmarks in numerous works [4, 13, 14].
> -	Different scales: small, medium, large. As the Reviewer 6qg1 said, the large-scale OGB datasets are challenging that many GNN based algorithms fail to generalize on large graphs.
> -	Different learning settings: we also conduct experiments on both transductive and inductive learning settings.
> -	Different tasks: we conduct experiments on two commonly tasks in graph learning: graph classification and node classification.
>
> We think the aforementioned experiments have already covered the common applications and settings in the graph learning field. In addition, we also conduct a comprehensive ablation study to demonstrate the impact of different components in AutoMasker and ICPG. In Figure 7 and Appendix, we also provide many insightful visualizations to illustrate rationality.  It is also appreciated by Reviewer 6qg1: "It is a good visualization of the kind of pruning the method is performing".
>
> **(2) For "not thorough solution" and "incorporating neighborhood information in the mask generation process".** We respectfully disagree with these two comments. We think you have missed the important information (GNN-based encoder) in our paper. I believe if you read our above responses carefully, your concerns will be alleviated.
>
> **(3) For "existing pruning baselines"** Please refer to **[Con 2: Stronger baselines]** above.
>
> [13] Benchmarking graph neural networks.
>
> [14] How Powerful are Graph Neural Networks?, ICLR'2019

---

> ### Author Response · Authors · 2021-11-23
> **We sincerely hope to discuss with you !**
>
> Dear Reviewer HhcH,
>
> Maybe you have missed some important information in our paper! For example, we adopt the GNN-based encoder in AutoMasker, not just MLP and sigmoid function. The GNN-based network will incorporate the topological information of the entire graph into node representations. So it can make decision based on a global view.
>
> We really hope to have a further discussion with you to see if our response solves your concerns.
>
> We would sincerely appreciate it if Reviewer HhcH could reply to the most important points in our rebuttal. For example, the insights about the paper in **[Con 1]**, stronger baselines in **[Con 2]**, the answers of the six questions in **[Con 3]**.
>
> We genuinely hope Reviewer HhcH could kindly check our response. Thank you!
>
> Best wishes,
>
> Authors

---

> > ### Comment · Reviewer_HhcH · 2021-11-27
> > **Thanks for the clarifications**
> >
> > Thank you for the clarifications. I have gone through all the reviews and your responses. Here are some questions I have.
> > 1. You claim that UGS uses mask based pruning which doubles the number of parameters. How is this different from the scores s_G that ICPG learns to decide which edges to keep and which to remove?
> > 2. One of the main sources of gains is the global view where a new embedding h is learned for all nodes using a GNN. Do you measure the gains from this strategy? In particular, if you remove the GNN encoder and use the features x directly, how is the performance affected? Is this what is plotted in Fig 6 (left) for the MLP strategy?
> > 3. The pruning in ICPG has multiple components (GNN encoder, MLP, followed by a new GNN). I imagine its training time will be higher than other baselines. Have you recorded the training time and compared it to the training time of other methods? This would present a training time vs. accuracy tradeoff to readers (I am not suggesting to do this during the rebuttal phase, only asking to share it if its available).

---

> > > ### Author Response · Authors · 2021-11-28
> > > **Response to Reviewer HhcH**
> > >
> > > Thank you very much for your reply! We are very happy to answer all your questions. We always believe that we will eliminate all your concerns to change your negative attitude towards our work. The following is our latest response:
> > >
> > > **1. Difference from UGS.** UGS is cumbersome because it assigns trainable parameters to all things. For the model, UGS assigns a mask to each weight, which will double the model parameters. For data, UGS assigns a mask to each edge, so that the trainable parameters are proportional to the number of edges in the graph. Therefore, when the number of edges in the graph data is very large (e.g. millions to billions of edges), the trainable parameters also become very large. We have reproduced their implementation on large-scale graphs and it usually causes the OOM issue in the training stage. Different from them, we adopt a completely different idea to reduce the computational cost and memory consumption:
> > > -	For the model, our method decides to prune the model based on the weight magnitude, avoiding the double-use mask similar to UGS.
> > > -	For data, we use AutoMasker to predict the score for each edge. The score matrix $\mathbf{s}_{\mathcal{G}}$ is **predicted  by AutoMasker rather than the trainable parameters (in UGS).** In this way, when the number of the graph's edges becomes very large, we can still use AutoMasker's invariable number of parameters to score all edges flexibly instead of adding more trainable parameters.
> > >
> > > **2. The gains from the global view.** Yes, you are right! We use a GNN encoder to incorporate the neighborhood information in the mask generation process, so that pruning is achieved from a global view. The benefits of this strategy are very obvious. We list the following two points to address your concerns:
> > > -	**Incorporating the neighborhood information is necessary.** From the result in Fig 6, "MLP" denotes replacing the GNN encoder network with an MLP network and keeping all the other parts unchanged. Since the GNN encoder uses 3 layers, for a fair comparison, we also adopt a 3-layer MLP network (with the same number of parameters). We conduct these experiments to claim the message-passing scheme of GNN is necessary for graph pruning. As a comparison, the MLP encoder network cannot incorporate the neighborhood information in the mask generation process so that it achieves poor performance.
> > > -	**Learning node representation is necessary.** You mentioned, "remove the GNN encoder and use the features x directly".  To alleviate your concerns, we remove the encoder network in AutoMasker and just use the node feature $\mathrm{X}$ (concatenation+MLP+ sigmoid) to conduct the experiments. The experimental results are shown in Table S2.  We can find that it (None) performs worse than GNN or MLP-based encoder network.  It shows that it is important to use a neural network to learn the node representation.
> > >
> > > Table S2. Test Accuracy (%) of graph classification and node classification with different encoder networks on different sparsity levels (14.26%, 22.62%, 33.66%, 40.13%).
> > >
> > > | Encoder Network       | RED-M5K (14.26%, 22.62%, 33.66%, 40.13%) | PPI (14.26%, 22.62%, 33.66%, 40.13%) |
> > > | :----------- | :------: | :------: |
> > > | GCN    |   (57.87±2.75, 57.69±3.64, 57.07±2.49, 56.63±2.97)  |    (98.69, 98.40, 97.82, 96.76)  |
> > > | MLP    |   (57.37±2.92, 56.49±3.61, 56.13±2.35, 55.59±3.13)  |    (98.14, 97.32, 96.17, 93.22)  |
> > > | None   |   (57.17±1.93, 56.19±3.17, 56.16±2.07, 54.87±2.34)  |    (98.07, 97.17, 95.42, 92.41)  |
> > >
> > > **3. Training time.** Although we use an additional GNN for co-training, our method is superior to UGS in terms of training time and computational cost. We give the following reasons:
> > > -	**Lightweight GNN encoder.** Our GNN encoder is often a small and lightweight model. For example, on the TU dataset, the hidden layer dimension of the GNN encoder is half of the GNN model. For Core node classification, the hidden layer dimension of the GNN encoder is a quarter of the GNN model (listed in Appendix).  However, UGS provides a mask for each parameter of the GNN model, so that the number of trainable parameters is doubled.
> > > -	**The score matrix $\mathbf{s}_{\mathcal{G}}$ does not need to be trained.** UGS provides learnable masks for both model parameters and graph edges, which further increases the training time and computational cost. However, our method performs pruning based on predicted scores. The score matrix $\mathbf{s}_{\mathcal{G}}$ is not a trainable parameter, it is only used as an indicator in the pruning stage.  In this way, our method can further save computational cost and training time.

---

> > > ### Author Response · Authors · 2021-11-28
> > > **(Continued) Response to Reviewer HhcH**
> > >
> > > **4. We are glad to show the training time comparison to readers.**
> > >
> > > In summary, compared with UGS, our method exponentially reduces the number of trainable parameters and computational cost in the training stage. To further solve your concerns, we evaluated the training time of our method on node classification and compared it with UGS. The results of node classification are shown in Table S3. It can be seen from the results that our method achieves less training time and keep a large gap (↓69.60%-72.94%) with UGS.
> > >
> > > Table S3. Average training time (s) of each sparsity level for graph lottery tickets learning.
> > >
> > > | Method      | Cora | Citeseer | Pubmed |
> > > | :----------- | :------: | :------: | :------: |
> > > | UGS    |   11.48  | 23.92  | 36.40 |
> > > | Ours   |   3.49   |  7.01  | 9.85  |
> > > | Time reduction   |   ↓69.60%  |  ↓70.69%  |  ↓72.94% |

---

> > > > ### Comment · Reviewer_HhcH · 2021-12-03
> > > > **Thank you**
> > > >
> > > > Thank you for addressing my concerns.

---

> > > > > ### Author Response · Authors · 2021-12-04
> > > > > **Thanks for your response!**
> > > > >
> > > > > Dear Reviewer HhcH,
> > > > >
> > > > > Thanks for your response! We are happy to answer any additional questions and provide more information.
> > > > >
> > > > > Best wishes,
> > > > >
> > > > > Authors

---

> > > ### Author Response · Authors · 2021-12-01
> > > **Looking foward to your reply!**
> > >
> > > Dear Reviewer HhcH,
> > >
> > > We look forward to your reply!
> > > We are glad that you understand our main contribution.
> > > According to your new concerns, we have gave our detailed point-to-point new response. We believe you will change your negative attitude towards our hard work.
> > >
> > > Best wishes,
> > >
> > > Authors

---

> > > ### Author Response · Authors · 2021-12-01
> > > **Sincerely expecting further discussions from Reviewer HhcH**
> > >
> > > Dear Reviewer HhcH,
> > >
> > > We really appreciated your time and your constructive reviews. We politely send you a kind reminder that the discussion period is ending within **24 hours**.
> > >
> > > Given our detailed replies and new experiments, could you please kindly check our response to see if it solves your concerns, so that you could also support our acceptance? Your support is very important to us and we greatly appreciate that!
> > >
> > > Meantime, please do not hesitate to reach out to us if there are other clarifications or experiments we can provide. Many thanks!
> > >
> > > Best Regards,
> > >
> > > Authors

---

> > > ### Author Response · Authors · 2021-12-02
> > > **Sincerely expecting further discussions from Reviewer HhcH**
> > >
> > > Dear Reviewer HhcH,
> > >
> > > We know that you may be very busy, and we are sorry to disturb you. But we sincerely hope you can take just a little time to read our response. This is very important to us!
> > > Please kindly let us know if there is anything else we can address to convince you for upgrading the scores.
> > > Thank you very very much!
> > >
> > > Best wishes,
> > >
> > > Authors

---

> > > ### Author Response · Authors · 2021-12-03
> > > **Your support is very important to us!**
> > >
> > > Dear Reviewer HhcH,
> > >
> > > Sorry to disturb you again, we have been waiting for you for a long time. Please kindly let us know if there is anything else we can address to convince you for upgrading the scores.
> > >
> > > Best wishes,
> > >
> > > Authors

---

> ### Author Response · Authors · 2021-11-25
> **Looking foward to your reply!**
>
> Dear Reviewer HhcH,
>
> Maybe you have missed some important information in our paper! We sincerely hope to discuss with you.
>
> Best wishes,
>
> Authors

---

> ### Author Response · Authors · 2021-12-02
> **Your support is very important to us!**
>
> Dear Reviewer HhcH,
>
> Sorry to disturb you, but we have been waiting for you for a long time. Please kindly let us know if there is anything else we can address to convince you for upgrading the scores.
>
> Best wishes,
>
> Authors

---

### Official Review · Reviewer_6qg1 · 2021-11-07

**Correctness:** 3
**Technical Novelty And Significance:** 3
**Empirical Novelty And Significance:** 3
**Recommendation:** 6
**Confidence:** 3

**Main Review:**

In Section 4, the authors evaluate ICPG on an extensive collection of datasets covering a several domains. Since the authors don't have a comparable baseline from previous works, they compare their method against Random Pruning which I think is acceptable.

ICPG consistently produces sparser graphs with higher accuracy than RP. The ability of ICPG to generalize on larger graphs compared to RP on the OGB dataset is a good result, many GNN based algorithms fail to generalize on large graphs.

A small comment on presentation - the x-axis in Figure 2 is not very readable.

In Section 4.4, the authors show that ICPG can indeed generalized to unseen graphs. This is a novel contribution. In Section 4.3, while the performance of ICPG drops faster than UGS, this is expected since UGS is not designed to generalize to unseen graphs. It is impressive that ICPG beats UGS while being able to generalize to unseen graphs.

Figure 7 is a good visualization of the kind of pruning the method is performing.

UPDATE: I acknowledge that I have read the author's response and maintain my score.


**Summary Of The Paper:**

The authors introduce ICPG, a novel method for pruning in graph and node classification tasks. The technique achieves good accuracy and is able to generate sparse graphs. Further, unlike previous work, the method is able to generalize to unseen graphs.

**Summary Of The Review:**

The authors introduce a novel way to prune graphs graph and node classification tasks which can generalize to unseen graphs. They evaluate the method on an extensive number of tasks. The method shows good performance.

Since there is no equivalent technique to compare against, the authors compare against random pruning which is probably a very weak baseline.

---

> ### Author Response · Authors · 2021-11-13
> **Response to Reviewer 6qg1**
>
> Thank you for your time, we are very glad that you have a positive impression on our work. According to your concerns, we provide the following responses:
>
> **[Con 1: The comparisons with other stronger baselines]**
> We have compared with UGS [1] on transductive node classification task. To our best knowledge, we are the first to achieve graph pruning on inductive graph learning, such as graph classification. The state-of-the-art graph pruning methods [1,2,3] are specially designed for transductive graph learning tasks, such as semi-supervised node classification on a single graph. Due to their inflexibility and implementation limitations, they are difficult to develop in inductive learning settings, such as graph classification. To further address your concerns, we adopt the sampling-based method GraphSAGE [4] and data augmentation-based method DropEdge [5], which are also designed for graph sparsification. For a fair comparison, we adjust the hyper-parameters in GraphSAGE  to achieve similar sparsity-levels as DropEdge and our method. The results are shown in Table S1. We can find that our methods consistently outperform other baselines at all sparsity-levels. It further demonstrates the superiority of our method.
>
>
>
> **[Con 2: About the presentation]**
> Thanks again for your carefully reading, we have accepted your valuable suggestions and updated the paper to make the x-axis in Figure 2 more readable.
>
>
>
> Table S1. Test Accuracy (%) of graph classification across different sparsity-levels. We perform 10-fold cross-validation to evaluate the performance, and report the mean accuracy.  Improvements denote the improved accuracy compared with the best baseline.
>
> | Method       | NCI1 (14.26%, 22.62%, 33.66%, 40.13%) | COLLAB  (14.26%, 22.62%, 33.66%, 40.13%)  | RED-B  (14.26%, 22.62%, 33.66%, 40.13%)  | RED-M5K  (14.26%, 22.62%, 33.66%, 40.13%)  |
> | :----------- | :-----------------------------------------------: | :-------------------------------------------------: | :------------------------------------------------: | :--------------------------------------------------: |
> | GraphSAGE    |           (77.79, 76.62, 72.97, 71.27)            |            (75.48, 73.93, 69.04, 68.50)             |            (84.32, 79.49, 79.35, 72.15)            |             (50.10, 47.65, 44.25, 36.83)             |
> | DropEdge     |           (83.09, 82.24, 81.40, 80.14)            |            (82.91, 82.16, 81.96, 81.52)             |            (91.40, 89.60, 88.00, 87.40)            |             (52.47, 50.37, 46.85, 45.35)             |
> | Our Method   |           (83.28, 82.82, 81.63, 80.34)            |            (83.68, 83.34, 82.90, 82.44)             |            (92.45, 92.15, 90.95, 90.10)            |             (57.87, 57.69, 57.07, 56.63)             |
> | Improvements |           (↑0.19, ↑0.58,  ↑0.23, ↑0.20)           |            (↑0.77, ↑1.18, ↑0.94, ↑0.92)             |            (↑1.05, ↑2.55, ↑2.95, ↑2.7)             |            (↑5.40, ↑7.32, ↑10.22, ↑11.28)            |
>
> [1] A unified lottery ticket hypothesis for graph neural networks, ICML'2021
>
> [2] Sgcn: A graph sparsifier based on graph convolutional networks, PAKDD'2020
>
> [3] GEBT: Drawing Early-Bird Tickets in Graph Convolutional Network Training
>
> [4] Inductive representation learning on large graphs, NeurIPS'2017
>
> [5] Dropedge: Towards deep graph convolutional networks on node classification, ICLR'2020

---

> ### Author Response · Authors · 2021-12-04
> **Looking foward to your reply!**
>
> Dear Reviewer 6qg1,
>
> Thank you for your reviews. We want to know if our response address your concerns. Please kindly let us know if there is anything else we can address to convince you for upgrading the scores.
>
> Best wishes,
>
> Authors

---

### Official Review · Reviewer_FXPK · 2021-11-14

**Correctness:** 3
**Technical Novelty And Significance:** 2
**Empirical Novelty And Significance:** Not applicable
**Recommendation:** 5
**Confidence:** 4

**Main Review:**

1) What is the practical purpose/use cases of the proposed method?  can the proposed method reduce the training time over GCN? Speeding up the inference? if so how much inference/training time can be saved? Or can the proposed method improve the prediction accuracy?

2) The paper seems getting confused about graph sampling over graph sparsity. See the section of "Graph Sparsification" in page 3. In fact, graphsage and fastGCN are both graph sampling method (definitely not for graph sparsity) aiming for speed up the training of GCN, and used in inductive setting, and is NOT transductive only.

3) what is the time complexity/running time for the proposed method? It seems to me that the proposed method is quite expensive---many rounds of GCN training and heavy computation over scoring the edge, # of edges of pair of data points. And how about  the scalability of the proposed method. I saw the experiment are all done on the small/median scale datasets.

4) What is the novelty of the proposed method of weight pruning? To my understanding, weight pruning using magnitude of weights score is quite well known--because there will be some redundancy in the weights during training.


**Summary Of The Paper:**

This paper is on generalizing UGS, (Chen et al., 2021) from transductive setting to inductive setting. The main idea is to learn the importance of each edge using the MLP over the learnt embedding from GNN, and pruning them through sorting the importance score. And weight pruning is done just by sorting the magnitude of weights in GCN.




**Summary Of The Review:**

I have concerns about novelty, scalability, and practical use case of the proposed method.

------ after reading the rebuttal

Thanks for the replies from the authors. I have read the reviews carefully. Some of my concerns are indeed solved, but I am still not convinced by the novelty, scalability and usefulness of the proposed method. For example, the use of this method is improving the inference time. The inference speed ups as shown in the comments are not significant, while lots of state-of-the-art inference speed up methods can do at least 10x speedup. Also according to the comments, the scalability is indeed a problem due to the lottery tickets problem, which is the main idea of the paper. So I think this paper's use scope is limited.

---

> ### Author Response · Authors · 2021-11-14
> **Response to Reviewer FXPK (Part of [Con 1])**
>
> Thanks for your reviews. It seems that you have a negative attitude toward our work. To address your concerns, we give the following point-to-point responses. We believe if you read it carefully, you will change your attitude.
>
> **[Con 1. Practical purpose/use and novelty]** These have been fully discussed in the introduction. Here we list the following points to make you more clear.
>
> **(1) For practical purposes.** The work is mainly aimed at a proof-of-concept study for extending the lottery ticket hypothesis [1] to wider graph learning applications. Although not targeted at a specific application, we find our proposed techniques to be practically relevant in several situations.
>
> - **Greatly reduce the computational cost in real-life applications.**  Graph lottery tickets will be suitable if a large graph (e.g., social network.) and its GNN model are deployed as a service on the cloud, and users could repeatedly query this graph. The query time will be largely shortened in the average sense. Furthermore, due to its inductive nature, it is also available for the dynamic network.
> - **GNN pre-training.** Graph-level transferability opens a new direction for GNN pre-training [2]: we can adopt the general knowledge learned from upstream tasks and transfer the knowledge to downstream tasks for efficient GNN training. It achieves the double-win: saving large computational costs and achieving better performance.
> - **GNN-level transferability.** We also find that the pruned subgraphs have excellent GNN-level transferability. It indicates that we successfully capture the critical subgraphs, which carry the significant semantic information.
> - **Inspirations for other research lines**. Firstly, numerous works are making efforts to refine and purify the noisy graph structure to achieve good robustness and generalization. Due to their transductive nature, there exists a large gap to directly transfer the optimization algorithms to inductive learning setting, which is significant for the graph learning field. The predictive model AutoMasker provides an idea to extend this research line. Secondly, the captured critical subgraphs always include the most significant information about the graph data. The wonderful visualizations provided in Figure 7 also give us strong confidence. The also provides more insights for graph explainability [3-5].
>
> **(2) For the novelty and contributions.**
>
> - **For method.** Existing graph pruning methods [6-8] mainly focus on a single graph with transductive learning settings, such as node classification on the Cora dataset. Various optimization methods can be easily applied to a given graph. However, in inductive learning settings, there exist numerous graphs with different nodes, which leads to different sizes of adjacency matrices. It is difficult for the current methods to deal with these irregular matrices. To overcome the dilemma, we use the AutoMasker to universally project and align diverse graphs into the same latent space for optimization, thereby avoiding directly optimizing various adjacency matrices with irregular sizes.
> - **For contributions.** To our best knowledge, we are the first to achieve data and model co-pruning for inductive graph learning. We successfully extend lottery tickets to a wider range of graph learning applications. Specifically, the proposed framework can flexibly develop on graphs with different types (molecules, superpixel graphs, social networks, citation networks), different scales (from small-scale to large-scale), diverse learning settings (inductive, transductive), and various tasks (node, graph classification). It proves that the proposed framework is more general, universal and applicable to many fields in graph learning.
> - **For other comments.** Finally, our novelty and contributions were also thankfully appreciated by Reviewer 6qg1 and Reviewer 4F4b, commented as “novel”, “impressive” and “significant meaning”.  As Reviewer 4F4b said: “It is very important to extend the concept of methodologies for the transductive setting to that of the inductive setting in GNNs.” As Reviewer 6qg1 said: “many GNN based algorithms fail to generalize on large graphs.”

---

> ### Author Response · Authors · 2021-11-14
> **(Continued) Response to Reviewer FXPK (Another part of [Con 1])**
>
> **(Continued) [Con 1. Practical purpose/use and novelty]**
>
> **(3) For reducing the computational cost.** Since the training of GNN on the graph follows the message-passing paradigm, the over-dense graphs and over-parameterized model lead to explosive memory consumption and computational cost. Our proposed pruning framework can significantly reduce the computational cost. Specifically, we conduct the experiment from small-scale to large-scale datasets to show the detailed inference cost. Following the work of UGS [1], we calculated MACs for better comparison. The experimental results are shown in Table S1. Compared with the baseline, our method can significantly reduce the computational cost by about 50%-90%, which shows the effectiveness of our method.
>
> **(4) For improving the prediction accuracy.** Our main contribution is to explore how to reduce the computational cost in the training and inference stage as much as possible while ensuring satisfactory performance. We do not care how to improve the prediction accuracy in this paper. In addition, we find that our method can also improve the prediction accuracy from two perspectives.
>
> - **AutoMasker can be regarded as a noise filter.** We found that the proposed AutoMasker can learn how to extract critical subgraphs, which can effectively prune the redundant or noisy edges. These subgraphs filtered by AutoMasker are more distinguishable, which can help various GNN models not to be interfered by noise edges and improve the performance. The specific experimental results are shown in Table S2.
> - **GNN-pre-training.** In section 4.4, we find that our method can benefit from pre-training: we can adopt the general knowledge learned from upstream tasks and transfer the knowledge to downstream tasks for efficient GNN training. It achieves the double-win: saving large computational costs and achieving better performance. The specific experimental results are shown in Table 1 in original paper.
>
>
>
> Table S1. Inference MACs of graph classification. Baseline denotes the full model with full graphs. Reduced computational costs are shown in brackets.
>
> | Method                    |      MUTAG      |       NCI1        |       COLLAB       |       RED-B        |       RED-5K       |      RED-12K       |    ogbg-code2     |      ogbg-ppa       |
> | :------------------------ | :-------------: | :---------------: | :----------------: | :----------------: | :----------------: | :----------------: | :---------------: | :-----------------: |
> | Inference MACs (Baseline) |     23.53M      |      834.97M      |      3445.43M      |      4723.60M      |     13661.66M      |     24366.16M      |     1397.95G      |      5680.79G       |
> | Inference MACs (Ours)     | 5.09M (-78.36%) | 223.76M (-73.20%) | 1103.06M (-67.98%) | 1583.97M (-66.47%) | 1584.23M (-88.40%) | 1533.57M (-93.71%) | 672.91G (-51.86%) | 2869.69G  (-49.48%) |
>
>
>
> Table S2. Test Accuracy (%) of graph classification. Ours: training and inference on the pruned sparse graphs (sparsity levels are shown in brackets). Baseline: training and inference with the original full graphs.
>
> | Method         |        NCI1        |        MUTAG        |       RED-5K        |       RED-12K       |
> | :------------- | :----------------: | :-----------------: | :-----------------: | :-----------------: |
> | GIN (Baseline) |     82.00±2.36     |     87.28±7.05      |     55.57±1.36      |     49.98±0.97      |
> | GIN (Ours)     | 83.07±1.65 (5.00%) | 89.88±6.54 (22.62%) | 56.67±2.06 (18.55%) | 50.46±2.59 (43.12%) |
> | Improvement    |       ↑ 1.07       |       ↑ 2.60        |       ↑ 1.10        |       ↑ 0.48        |
> | GAT (Baseline) |     82.11±2.05     |     89.42±7.85      |     56.21±1.62      |     49.40±1.09      |
> | GAT (Ours)     | 82.73±1.53 (5.00%) | 90.44±5.43 (18.55%) | 56.49±1.25 (9.75%)  | 49.77±3.44 (9.75%)  |
> | Improvement    |       ↑ 0.62       |       ↑ 1.02        |       ↑ 0.28        |       ↑ 0.37        |
>
>
> [1] The Lottery Ticket Hypothesis: Finding Sparse, Trainable Neural Networks, ICLR'2019
>
> [2] Graph contrastive learning with augmentations, NeurIPS'2020
>
> [3] Explainability in graph neural networks: A taxonomic survey.
>
> [4] Generative causal explanations for graph neural networks, ICML'2021
>
> [5] Gnnexplainer: Generating explanations for graph neural networks, NeurIPS'2019
>
> [6] A unified lottery ticket hypothesis for graph neural networks, ICML'2021
>
> [7] Sgcn: A graph sparsifier based on graph convolutional networks, PAKDD'2020
>
> [8] GEBT: Drawing Early-Bird Tickets in Graph Convolutional Network Training

---

> ### Author Response · Authors · 2021-11-14
> **(Continued) Response to Reviewer FXPK [Con 2] and [Con 3]**
>
> **[Con 2. Graph sampling v.s. Graph Sparsification.]**
>
> We do not confuse these two simple concepts and give the following points for this concern.
>
> **(1) "Sparsification" is a very general definition.** The aim of graph sparsification is to find small subgraphs from input large graphs that best preserve desired properties. Graph sampling is just a technique to sample subgraphs from the original graph in each training round. From your description, "Sparsification" seems to be a kind of pruning for graph data, that is, this subgraph remains unchanged throughout the training or testing stage. The sampling-based method [9, 10] still retains the complete graph data, but extracts its sub-parts in each training step. But in our opinion, the graph sampling method is just a new process to achieve graph sparsification in the training process. From a global view, the graph sampling method does use a new subgraph to train the model, and this subgraph retains the main attributes of the original graph. Therefore, we did not make such a detailed distinction between the concepts of the two when writing this paper.
>
> **(2) Views from other works.** While writing this paper, we have extensively investigated many works on graph pruning or sparsification. In these graph sparsification works [6, 7, 8, 11], the graph sampling-based methods are also mentioned.  For example, these two works [6, 11] clearly pointed out in related work that graph sampling can be used as a way to sparsify graphs. These two works [7, 8] even used GraphSAGE [9] as the baseline for comparison. Therefore, the mainstream view is that the graph sampling method can indeed be regarded as another kind of graph sparsification.
>
> **(3) We did not claim that GraphSAGE and FastGCN cannot apply to inductive learning.** We use the "Another research line" to separate the sampling-based methods and optimization-based methods. We claim that the optimization-based methods, such as SGCN [7], cannot apply to inductive learning setting.
>
> **(4) We have updated the draft.** Considering the above expressions may cause confusion, we respectfully accept your valuable suggestions and revise the related work. Thanks for your careful reading.
>
>
>
> [9] Inductive representation learning on large graphs, NeurIPS'2017
>
> [10] Fastgcn: fast learning with graph convolutional networks via importance sampling, ICLR'2018
>
> [11] Robust Graph Representation Learning via Neural Sparsification, ICML'2020
>
>
>
> **[Con 3. Time complexity and scalability]**
>
> **(1) Finding the lottery tickets efficiently is another research line！** Please carefully read the paper about the lottery ticket hypothesis (LTH) [1]. It points out that we can independently train a sparse network to achieve a similar performance level as the original dense network. This gives the possibility of saving computational costs before training. One of our contributions is to extend the lottery hypothesis to wider graph learning applications. It points that a similar phenomenon exists in the graph learning community. Multiple iterations of pruning ensure high-quality lottery tickets, which has been confirmed in many works [12-14]. How to find lottery tickets with a low computational cost is another research line. We list the following works for reference.
>
> -	**Explore the early stop mechanism.** Early-bird tickets [15] propose an early stop strategy to locate the lottery tickets at an earlier stage. It greatly saves the computational cost to find the lottery tickets.
> -	**Less data is more.** In our framework, we use all the training data to find the GLTs. Recent work [16] proposes a data-efficient ticket finding strategy to further save the computational cost.
> -	**Dynamic sparse training.** The work [17] propose a novel pruning framework to dynamically prune the network parameters in the training stage. It also greatly save computational cost and can achieve similar performance as the IMP [1].
>
> Obviously, we can borrow similar ideas to improve our method as our future work.
>
> **(2) About the scalability.** Please carefully recheck our paper in **Figure 3** (page 6) and Obs.4. again. We have conducted two experiments on the large-scale OGB dataset: ogbg-ppa and ogbg-code2. Furthermore, our large-scale experiments have been commented by Reviewer 6qg1 that, the large-scale OGB datasets are challenging that many GNN based algorithms fail to generalize on large graphs.
>
>
>
> [12] The lottery tickets hypothesis for supervised and self-supervised pre-training in computer vision models, CVPR'2021
>
> [13] The lottery ticket hypothesis for pre-trained bert networks, NeurIPS'2020
>
> [14] GANs can play lottery tickets too, ICLR'2021
>
> [15] Drawing early-bird tickets: Towards more efficient training of deep networks, ICLR'2020
>
> [16] Efficient lottery ticket finding : Less data is more, ICML'2021
>
> [17] Rigging the lottery : Making all tickets winners, ICML'2020

---

> ### Author Response · Authors · 2021-11-14
> **(Continued) Response to Reviewer FXPK [Con 4] and [Con 5]**
>
> **[Con 4. The novelty about the weight pruning.]**
>
> **(1) Magnitude-based pruning is not our main contribution.** As you said, magnitude-based pruning is already a very mature pruning algorithm, which has been used in many fields [12-16]. But in this paper, we hope you can focus on our main contribution: AutoMasker.  Model pruning aims to improve the performance of the AutoMasker. They can help each other and form a virtuous circle. We have fully discussed the importance of model pruning in ablation study. Model pruning can be seen as a kind of regularization to improve performance. It also prevents over-fitting. We do not want to emphasize  that the weight pruning method is our contribution, but want to emphasize the importance of co-pruning.
>
> **(2) Magnitude-based pruning can be seen as an improvement of the UGS framework.** UGS adopts mask-based pruning for the GNN model. This will double the trainable parameters and increase the computational cost, which we have fully discussed in the introduction. We revealed this shortcoming, improved this pruning strategy, and achieved better performance in our framework.
>
>
> **[Con 5. Responses to Summary Of The Review.]** We have provided detailed responses to all of your concerns. For novelty and practical use cases, we have prudently provided the responses in **[Con 1. Practical purpose/use and novelty]**. For scalability, you have missed our important results in Figure 3. Please carefully double-check the paper again. We believe you will withdraw your negative attitude if you read all the responses carefully. Thank you for your time again and looking forward to your reply!

---

> ### Author Response · Authors · 2021-11-23
> **We really hope to have a further discussion with you!**
>
> Dear Reviewer FXPK,
>
> We really hope to have a further discussion with you to see if our response solves your concerns.
>
> Maybe you have missed some important things in our paper.  We would sincerely appreciate it if Reviewer FXPK could reply to the most important points in our rebuttal. For example, the practical purpose in **[con 1]**, Graph sampling v.s. Graph Sparsification in **[con 2]**, time complexity and scalability in **[con 3]** and the novelty about the weight pruning in **[con 4]**.
>
> We genuinely hope Reviewer FXPK could kindly check our response. Thank you!
>
> Best wishes,
>
> Authors

---

> ### Author Response · Authors · 2021-11-30
> **Looking foward to your reply!**
>
> Dear Reviewer FXPK,
>
> We sincerely hope we can have a further discussion with you!
> We genuinely hope Reviewer FXPK could kindly check our response. Thank you
>
> Best wishes,
>
> Authors

---

> ### Author Response · Authors · 2021-12-01
> **New response to Reviewer FXPK.**
>
> Thank you very much for your reply and the improved score.  We give the following supplementary explanations to alleviate your concerns. We always believe we can change your negative attitude.
>
> **1. Novelty.** We believe our novelty is strong and we list the following supplementary points:
> -	**Data-model co-designed.** Our method is an end-to-end data-model co-designed pruning framework. Most of the inference acceleration methods only focus on how to prune the model, but we are the first to explore the data-model co-designed for inductive graph learning. The benefits of our proposed method not only come from the lightweight model, but also from the sparse graph data. For graphs with very dense edges, the gains of our method are very significant.
> -	**Irregular graphs are handled uniformly.**  The AutoMasker cleverly solves the issue of different graph sizes. We optimize the graph data by mapping irregular graphs to a unified space.
>
> **2. Scalability.** We respectfully disagree with your comments "*this paper's use scope is limited*". Our framework can easily develop on a wider range of graph learning applications and we have provided numerous experiments to prove this. Specifically, the proposed framework can flexibly develop on graphs with:
> -	**Different types**: molecules, superpixel graphs, social networks, citation networks;
> -	**Different scales**: from small-scale, medium-scale to **large-scale OGB datasets**;
> -	**Different learning settings**: inductive and transductive;
> -	**Different tasks**: node and graph classification;
>
> It proves that the proposed framework is more general, universal and applicable to many fields in graph learning.
>
> **3. Practicality.** We think you may have missed some important points on practicality. In addition to *GNN pre-training, GNN-level transferability, inspirations for other research lines* as we mentioned before, we also added the following points:
> -	**State-of-the-art inference speed-up methods cannot easily work on graph learning.**  In the GNN field, due to a large number of dense connections of graph data, the GNN model requires very frequent message-passing steps. Therefore, **the over-dense graph data is the bottleneck** that largely restricts the inference speed, but **few works pay attention to this point.** Although there exist works for accelerating DNN inference, **these methods cannot be well transferred to GNNs.** Our method makes efforts to break the bottleneck and achieve faster inference in the GNN field. This is a totally new application that accelerates inference from the data level. In addition, as you said: "*The inference speed ups as shown in the comments are not significant, while lots of state-of-the-art inference speed up methods can do at least 10x speedup*", but what we show in the table is the total inference MACs as work UGS. For inference time, our method can speed up the time by about 2x-15x.  This speed gain depends on the graph dataset. The denser the graph data, the greater the benefits we get.
>
> -	**Less memory consumption.** With graphs rapidly growing in size, the memory space required for inference has also shown explosive growth. The out-of-memory (OOM) problem seriously hinders the wide application of GNN. Although some methods can save the inference time, the OOM issue still exists due to the over-dense graphs and over-parameterized GNNs. We for the first time explore how to further reduce the memory consumption from the two orthogonal perspectives: data and model.  From our sparsity results, we can save about 31-67% of memory consumption. This effectively alleviates the OOD issue.
> -	**Lightweight graphs and models are important in practical application.** GLT will be suitable if a large graph and its GNN are deployed as a service on the cloud (e.g. AWS), and users could repeatedly query this graph, e.g., on different nodes. Then in an average sense, the query time per node might be shortened.
> -	**Potential to save training costs.** Although there exist methods that make efforts to reduce the inference time, no work tries to save the cost in the training stage. GLT tells us that we can train sparse graphs with sparse GNNs at the beginning to largely save the training costs.

---

### Author Response · Authors · 2021-11-20
**Looking forward to your reply**

Firstly, we thank all the reviewers' time for the review. We have already provided detailed responses. Since the first discussion period will end soon, we really hope to have a further discussion with reviewers to see if our responses solve your concerns. Your suggestions and comments are valuable to us.

---

### Author Response · Authors · 2021-12-01
**General Response**

Dear AC and all reviewers:

We really appreciate all the reviewers for their valuable suggestions. We are thankful for the reviewers appreciating our work’s solid contributions.

We are glad that reviewer **4F4b** and **6qg1** have positive ratings on our work.

We thank reviewer **FXPK** for constructive reviews and really appreciate reviewer **FXPK** for increasing our score.

We sincerely hope to have further discussions with reviewer **HhcH** to see if our response solves his/her concerns. Please kindly let us know if there is anything else we can address to convince you for upgrading the scores.

We are happy to answer any additional questions and provide more information.

---

### Decision · Program_Chairs · 2022-01-20

**Decision:**

Reject

**Comment:**

This paper studies the pruning problem of graph neural networks, i.e. finding lottery tickets for GNN. In particular, it generalizes UGS by Chen et al. (2021) from transductive setting to inductive setting where prediction on unseen graphs is possible. The main idea is: 1) learn a mask network to assign importance scores for edges using the embedding features of the nodes connected, that avoids the double parameter memory costs in UGS; 2) prune the edges according to the importance score and weights of GCN according to their magnitudes. Main concerns from reviewers are about the novelty, evaluation, and scalability. Despite that generalization to unseen graphs using the mask functions on embedding features is a new aspect, the evaluation is compared with relatively weak baselines and inference time scalability of is still an issue.